# Reconfigurable photonics with on-chip single-photon detectors

Samuel Gyger 1✉, Julien Zichi[1], Lucas Schweickert 1, Ali W. Elshaari 1, Stephan Steinhauer 1, Saimon F. Covre da Silva[2], Armando Rastelli 2, Val Zwiller 1, Klaus D. Jöns 1 & Carlos Errando-Herranz 1✉

Integrated quantum photonics offers a promising path to scale up quantum optics experiments by miniaturizing and stabilizing complex laboratory setups. Central elements of quantum integrated photonics are quantum emitters, memories, detectors, and reconfigurable photonic circuits. In particular, integrated detectors not only offer optical readout but, when interfaced with reconfigurable circuits, allow feedback and adaptive control, crucial for deterministic quantum teleportation, training of neural networks, and stabilization of complex circuits. However, the heat generated by thermally reconfigurable photonics is incompatible with heat-sensitive superconducting single-photon detectors, and thus their on-chip co-integration remains elusive. Here we show low-power microelectromechanical reconfiguration of integrated photonic circuits interfaced with superconducting single-photon detectors on the same chip. We demonstrate three key functionalities for photonic quantum technologies: 28 dB high-extinction routing of classical and quantum light, 90 dB high-dynamic range single-photon detection, and stabilization of optical excitation over 12 dB power variation. Our platform enables heat-load free reconfigurable linear optics and adaptive control, critical for quantum state preparation and quantum logic in large-scale quantum photonics applications.

[1] Department of Applied Physics, KTH Royal Institute of Technology, Stockholm, Sweden. [2] Institute of Semiconductor and Solid State Physics, Johannes Kepler University Linz, Linz, Austria. ✉email: gyger@kth.se; carloseh@kth.se

Optical quantum technologies are crucial to materialize the promises of quantum communication[1], quantum computing[2], and quantum simulation[3]. These applications require a leap in system complexity, only achievable via the miniaturization and stability provided by large-scale photonic integrated circuits (PICs)[4].

A quantum PIC is formed by a set of building blocks such as single-photon sources, quantum memories, reconfigurable photonic circuits, and detectors[5–7]. Reconfigurable photonic circuits not only provide the link between the other building blocks, but also enable the linear optic operations required for quantum state preparation and quantum logic[8]. In particular, combining reconfigurable photonics with detectors is central for on-chip single-photon detection and to enable feedback and adaptive control. Feedback is essential for quantum communication and computation protocols based on deterministic teleportation[9], for self-configuration of arbitrary linear optics[10], and for monitoring and stabilization of power, phase, and polarization. Elements addressing these functions usually outnumber other devices in proposed protocols and experimental setups, and thus their on-chip integration is a central challenge, often overlooked, towards the upscaling of classical and quantum optics[11,12]. This requires reconfigurable elements with low optical loss, a small footprint, and low electrical power consumption for cryogenic compatibility. Traditional PIC reconfiguration based on thermo-optic[13], carrier dispersion[14], and electro-optic $\chi^{(2)}$ effects[15] suffers from high power consumption, high optical loss, and large footprint respectively. A promising cryogenic compatible reconfiguration method is microelectromechanical (MEMS) actuation, which combines low power consumption, low optical loss, and small footprint[16]. However, to date, there has been no demonstration of the compatibility of reconfigurable photonics with single-photon detectors in the same quantum PIC[17].

Here, we integrate MEMS PIC reconfiguration with superconducting single-photon detection on the same chip, and show three crucial components of quantum optics experiments. We demonstrate reconfigurable routing of classical light and single photons, high-dynamic range detection of optical excitation powers and single photons, and power stabilization of optical excitation using a feedback loop.

## Results

**Waveguide-coupled single-photon detectors**. Superconducting nanowire single-photon detectors (SNSPDs) provide broadband detection of single photons with high detection efficiency, high signal-to-noise ratio, fast recovery time, and low timing uncertainty[18]. Their compatibility with mature PIC material platforms[19] and their excellent on-waveguide performance makes them outstanding integrated single-photon detectors[7].

In this work we fabricate hairpin[20] SNSPDs from a 9-nm-thin NbTiN film[21] on top of $Si_3N_4$ waveguides (see Fig. 1a, and find a description of the sample and fabrication process in Methods section and in Supplementary sections I and II). The 90-nm wide and 40-μm long photo-sensitive part of the wire is connected in series to a lumped-element inductor that prevents latching[22]. We simulated the absorption of our hairpin detectors to be 95.5% (see Supplementary section III). The detectors exhibit a saturated detection regime at 795 nm wavelength (measurement setup description in Supplementary section IV), revealed by the sigmoidal shape of the detection efficiency versus the bias current (Fig. 1c), which indicates unity internal quantum efficiency[23]. The two detectors feature critical currents of 15.8 μA (Detector A) and 5.9 μA (Detector B). The difference in critical current between dark and illuminated measurement is linked to the stochastic

nature of the superconducting to normal state transition, leading to a spread in the measured switching currents[24]. The detectors show different on-chip detection efficiencies (Detector A is 44.6 times more efficient than Detector B), which we attribute to a lithographic defect in Detector B, which results in a nanowire constriction and lower current densities, and therefore lower detection efficiency in the remaining nanowire (see Fig. S2)[25]. The devices show timing jitters of 121.0(19) ps and 253.0(14) ps with room temperature amplification, and reset times with an exponential decay of 4.73(3) ns and 4.71(1) ns (detectors A and B respectively, see Supplementary section IV).

**Low-power microelectromechanics with superconducting detectors**. Here, we demonstrate capacitive MEMS tuning as an SNSPD-compatible reconfiguration mechanism suitable for large-scale quantum PICs. Figure 1a shows a schematic of our device, and Fig. 1b shows a scanning electron microscope (SEM) image highlighting the two grating couplers for coupling of input light and part of the MEMS-tunable beam splitter. More information on our sample and SEM images of the other parts of the device, including the two output SNSPDs can be found in the Supplementary section I and Figs. S1 and S2.

We use the same NbTiN layer to build the MEMS actuators, electrical connections, contact pads, and single-photon detectors. Our fabrication process, described in the Supplementary section II, is largely enabled by the etch-resistance of NbTiN to hydrofluoric acid. The MEMS actuator consists of a NbTiN-on-$Si_3N_4$ cantilever, suspended over the Si substrate. The application of an electrical potential between the cantilever and the substrate forms a charged capacitor, which is subject to an attractive force that bends the cantilever vertically. The cantilever is attached to one of the two air-clad waveguides forming a directional coupler, and actuation results in an increase of the vertical separation between the waveguides, as illustrated in Fig. 1a. This, in turn, reduces the modal overlap, and thus changes the splitting ratio of the beam splitter, which we measure using the integrated SNSPDs (a theoretical description of our device and simulated actuation curves can be found in Supplementary section V). The measured tuning curve is shown in Fig. 1d, and follows our simulations (see Supplementary section V), yielding a high extinction ratio of 28.1 dB. The lower part of Fig. 1d shows the power ratio (defined as $PR = 10 \log_{10} \frac{counts_B}{counts_A}$) between detectors, and highlights the actuation voltage at which the highest extinction ratio was achieved. We observe an on–off ratio for individual ports of 27.5 dB (detector A) and 11.7 dB (detector B). We attribute the difference in on–off ratio to different detector efficiencies, as described in Supplementary section VI.

The normalized frequency response of our device is shown in Fig. 2a, where the amplitude $|A|(\omega)$ is defined as the normalized amplitude ($|A|(\omega)/|A|(\omega \to 0) = 1$) of the modulated SNSPD counts under a sinusoidal actuation voltage with amplitude $\Delta V(\omega)$ (see schematic in the inset of Fig. 2a, and find further description in the Supplementary section VII). We observe constant modulation amplitude up until the first mechanical resonance frequency between 1 MHz and 2 MHz, in line with our simulated value of 1.6 MHz (see Supplementary section V). Our device presents stable and reliable operation, with hysteresis below 2.4% and power stability with a standard deviation below 0.5% over 60 min (see Supplementary section VIII). During the frequency sweeps, we performed more than 20 million switching events proving the durability of integrated MEMS devices. To further demonstrate device robustness, we performed three cooldown cycles and confirmed the operation of the MEMS and detectors. In addition, the robustness of this geometry in terms

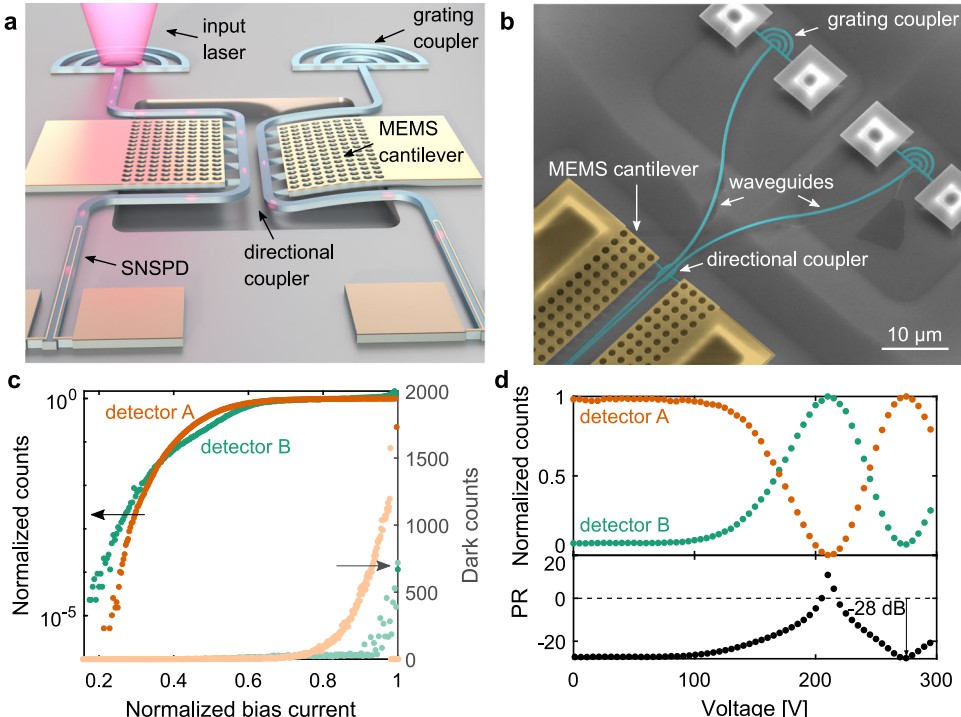

**Fig. 1 Device description and characterization. a** Artist view of the demonstrated device, composed of grating couplers for light input and a MEMS reconfigurable beam splitter connected to two superconducting single-photon detectors. **b** False-colored SEM of the input section of our device, showing the waveguides and grating couplers, and the MEMS actuator and electrodes. **c** Photon count rate at a wavelength of 795 nm for the two on-chip detectors and the corresponding dark counts, normalized to their individual saturated detection. The reduced critical current in the dark count measurement of detector A is due to measurement-to-measurement fluctuations in the critical current. **d** Measured photon detection counts versus MEMS actuation voltage, normalized to the individual maximum transmission and power ratio (PR) between detectors. Note that the detectors feature different detection efficiencies (44.6 times higher in A, see Results section).

of yield is confirmed by the operation of three copies of the same MEMS actuator on the same chip. In terms of power consumption, for a 100-kHz drive and full splitting ratio tuning, we estimate a power consumption below 75 µW, dominated by the capacitance of our contact, since the MEMS actuator consumes 2.24 µW under these conditions (see Supplementary section IX for power consumption derivation). This power, only consumed during dynamic actuation, is dissipated along the non-superconducting transmission line, which is limited to the off-chip components and in particular the high-resistivity voltage source, and thus far from our SNSPDs. Under DC actuation, the power consumption is driven by leakage currents. Due to the high insulation of vacuum and $SiO_2$ in our capacitor, leakage currents are minimal, which leads to power dissipation in the fW-range despite the high voltages applied. For example, for a 50:50 beam splitting ratio we estimate the power dissipation to be 6 fW, and for full inversion of the signal, 8.5 fW.

**An on-chip power meter with high dynamic range.** Although measuring the power in any optical setup is a mundane task, it is crucial for the setup, troubleshooting, and success of any measurement. While a macroscopic power meter can be found in most beam paths, PICs make access to the optical signal more challenging and require integrated photodetectors. In PICs, these are commonly built using complex processes such as heterogeneous integration of Ge or InGaAs on Si[26]. In contrast, we combine the two SNSPDs and the reconfigurable MEMS, fabricated with our simple fabrication process using the same NbTiN layer, to demonstrate a power meter with a linear dynamic range exceeding 90 dB and sensitivity down to the single-photon level.

Figure 2c shows three switchable regions of our high dynamic range power sensor that are connected by measuring detection counts while sweeping the MEMS actuation voltage. The first range is measured with most of the optical power routed into detector A. The second range is covered by detector A and detector B, where most of the optical power is routed to the lower efficiency detector B. The third range is covered by detector B with most of the optical power routed to detector A, thereby saturating detector A. The low efficiency of detector B, which enables this measurement, is attributed to fabrication variations and could be carefully engineered by varying the nanowire length coupled to the waveguide mode[27]. More information on the device function and our characterization setup can be found in Supplementary section X.

Our measurement range is in line with dedicated commercial devices and experimental demonstrations[28,29] but is unique in which it is sensitive at the single-photon regime, as we demonstrate in following sections. This high sensitivity to very low power levels makes our sensor ideal for tap-based circuit sensing where only a fraction of the light in the waveguide is available to the photodetector. This is crucial for built-in self-tests and wafer-level testing[30]. In the following paragraphs, we describe the use of this on-chip power sensor to generate direct feedback for power control in our circuit.

**On-chip power stabilization.** The large tunable extinction ratios and near-MHz speed of our device, combined with the on-chip detectors, enable dynamic stabilization of optical power in one of the beam splitter arms. Power stabilization and control is ubiquitous in classical and quantum optics experiments[31], with key

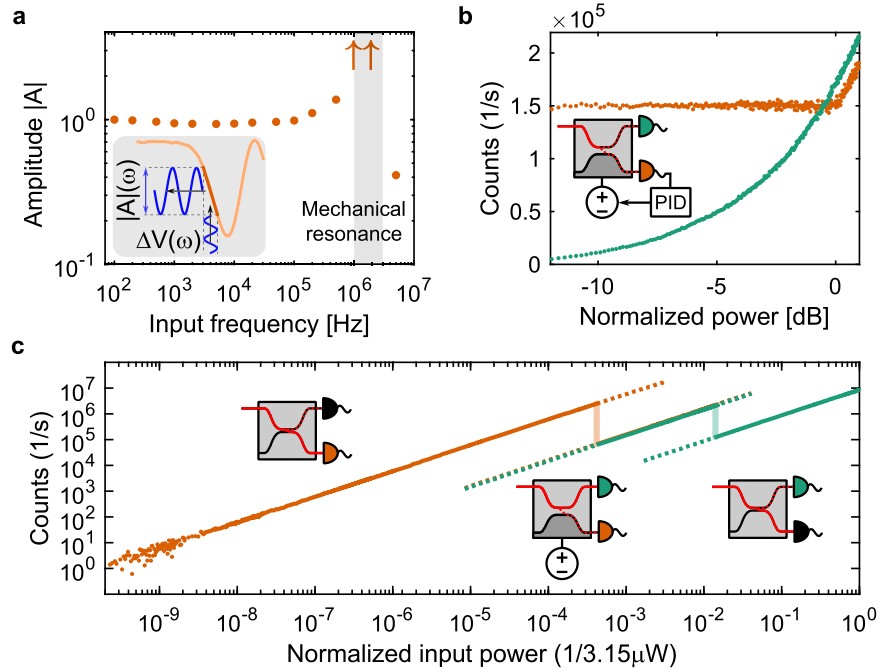

**Fig. 2 Frequency response and demonstrated applications. a** Frequency response of the tunable beam splitter measured using an on-chip detector up to the first resonance frequency normalized to the DC amplitude. The arrows represent the low bound of those measured amplitudes. Inset: this measurement was performed by translating the actuation voltage into a modulation of the splitting ratio. **b** PID-controlled power stabilization on one arm (detector A, orange) using the on-chip detectors and the MEMS-tunable beam splitter. The other arm (detector B, green) detects the rerouted power. The detection events are counted by the driving electronics of the detectors. **c** 90 dB dynamic range on-chip photodetector that combines a high and a low-efficiency detector with switchable measurement ranges. The vertical connections are the measured counts while changing the MEMS voltage from 0 V to 196.5 V. The insets show the active detectors and MEMS settings in each of the three ranges: for lowest input power, Detector A is used with most of the power routed to its waveguide. For medium input powers the MEMS splitter routes most of the optical power to lower-efficiency Detector B, and both detectors can be used. For highest optical power, no actuation is applied and low-efficiency Detector B is used.

examples being spectroscopy[32] and deterministic single-photon generation[33]. In current experiments, this is implemented off-chip and requires support hardware and laboratory space, which can be minimized or eliminated by using our on-chip device. While on-chip experiments present higher stability within the chip, the input coupling efficiency of an off-chip excitation laser is highly sensitive to polarization and mechanical movement. In a packaged photonic circuit with an on-chip light source, not only thermal fluctuations but also fabrication process variations and aging effects need to be compensated[34]. We address these problems by providing direct feedback on the MEMS beam-splitter using the power measured on the on-chip detector A, while the excess power is routed into the second arm (detector B). Figure 2b shows the measurement results and a schematic of our device. A PID feedback loop uses the detection counts provided by the SNSPD driver electronics to stabilize the measured power without manual intervention. The circuit yields stable (within 1.3% standard deviation) on-chip power while we tune the off-chip input power over 12 dB. The control loop was run every 100 ms and the input laser power was swept from 0 µW to 300 µW in 1 µW s$^{-1}$ steps. Details can be found in Supplementary section XI.

**On-chip reconfigurable single-photon detection**. To demonstrate the performance of our device in the single-photon regime for quantum PICs, we characterized it using single photons from an on-demand single-photon source. The source consists of a GaAs quantum dot excited via two-photon resonant excitation[35] with a 320-MHz repetition rate and a pulse length of 40 ps (see Supplementary section XII). Single photons from the exciton transition with an energy of 1.5636 eV (795 nm, spectrum in the

inset in Fig. 3a), are coupled via an optical fiber and free-space optics into our device using one of the grating couplers.

We performed a lifetime measurement on-chip, as seen in Fig. 3a, resulting in an exciton lifetime $\tau_X = 232(2)$ ps, in line with the off-chip measurement of $\tau_X = 216.3(6)$ ps. We extract the lifetime by fitting a decay convoluted with the instrument response function of the detector measured at the same wavelength using a 3-ps pulsed laser (see Supplementary section XII). We then measured single-photon purity by performing a Hanbury Brown and Twiss (HBT) measurement by routing the photons from our source into an off-chip beam splitter with one arm coupled through our device into the on-chip detector A, and the other arm going into a commercial off-chip SNSPD (measurement setup schematic in Fig. 3c). Figure 3b shows our measurement results, yielding a $g^{(2)}(0) = 0.11 \pm 0.05$, clearly showing anti-bunching and single-photon detection, in line with the $g^{(2)}(0) = 2.97 \times 10^{-4}$ measured using two commercial SNSPDs with the same emitter.

The discrepancy between our on-chip and off-chip $g^{(2)}(0)$ is due to the limited optical coupling efficiency into the chip leading to a degraded signal-to noise ratio between dark counts of the detector and detection events due to single photons. The optical coupling efficiency is limited by our grating coupler design, which was experimentally optimized based on the designs in Zhu et al.[36]. We believe that our longer wavelength (and thus lower refractive index contrast) and the non-optimized distance to the substrate limits our achievable coupling efficiencies, which can be potentially improved by fabricating additional grating periods. Additionally, the coupling efficiency can be improved via low-loss fiber coupling[37], or quantum emitter integration using monolithic[38] or hybrid approaches[17].

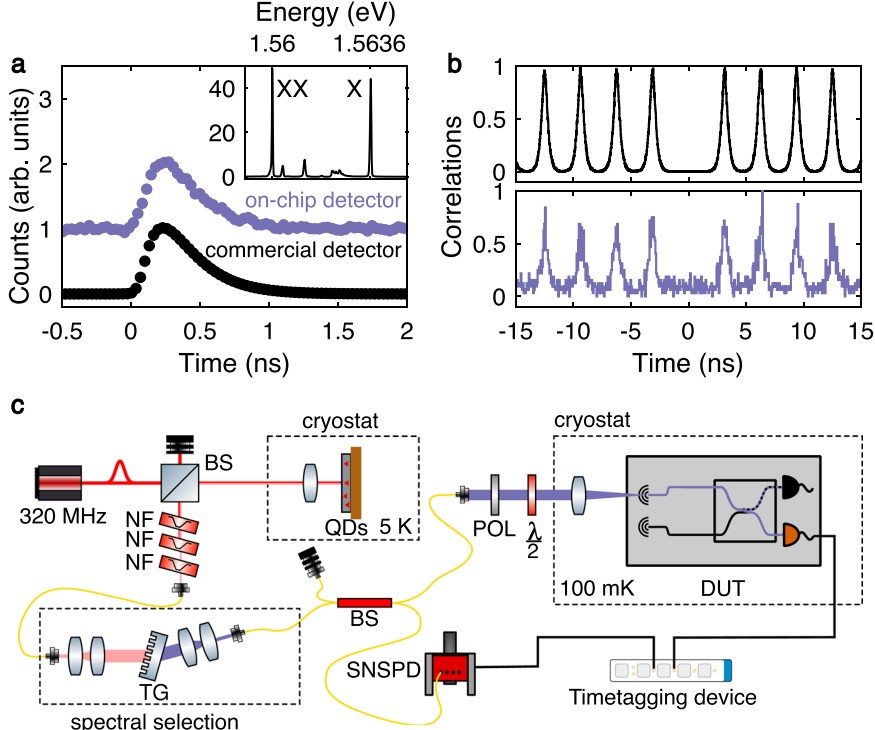

**Fig. 3 Single-photon experiments. a** Lifetime measurements using our device (purple) compared to commercial fiber–coupled SNSPDs (black). Inset: spectrum of the deterministically excited quantum dot used in this work, under $\pi$ pulse excitation, with highlighted exciton (X) and biexciton (XX) lines. **b** HBT measurements, showing a comparison of detection off-chip (top) and detection with one SNSPD on-chip (bottom). **c** Sketch outlining the experimental configuration. For **a** the lifetime of the QD is either fully measured on-chip or using the commercial SNSPD system. For **b** one arm of the HBT setup is on-chip while the second arm is connected to the SNSPD system. The coincidences are then measured using a time-to-digital converter. Optical components: BS beam splitter, NF notch filter, TG transmission grating, POL polarizer, ($\frac{\lambda}{2}$) half waveplate, DUT device under test.

## Discussion

Our proof-of-principle implementation of this technology could benefit from established fabrication processes and reach state-of-the-art performance. For example, improved SNSPD designs and readout electronics have demonstrated longer wavelength detection[20], shorter reset times[39], and sub-3-ps detection jitter[40]. While the number of devices in our proof of concept experiment is too small to estimate a yield (see Supplementary section II), in an industrial fabrication process, fabrication yield is expected to be no limiting factor[41,42]. For high frequency and high voltage signals ($\approx$150 V and >10 kHz), our MEMS actuation signal creates detection events on the detector channels even on unbiased detectors due to the large amplification in the readout channel (see Supplementary section VII for a more in-depth discussion). This can be improved by better electrode design on-chip and by reducing the actuation voltage.

MEMS technology is inherently scalable and reliable, as evidenced by the success of MEMS sensors in commercial electronics. The high voltage actuation required by our device is due to (1) the large capacitive gap between device layer and substrate and (2) the out-of-plane bending of the device caused by stress relaxation on the $Si_3N_4$ and NbTiN layers, which requires stiff cantilevers to maintain evanescent coupling between the directional coupler waveguides. We believe that this is not a fundamental limit in our platform, the actuation voltage can be reduced by tuning the stack stress distribution, by designing strain-tolerant MEMS actuators, and by using in-plane actuators with smaller capacitive gaps[16]. For example, capacitive MEMS in silicon have demonstrated sub-10-V actuation[16] and excellent scalability[43], as evidenced by the demonstration of a $240 \times 240$

photonic MEMS switch matrix, i.e., the largest silicon photonic circuit demonstrated to date[44]. These features make it likely that foundries will offer photonic MEMS soon, and our platform provides a clear pathway towards simultaneous integration of MEMS and SNSPDs in a simple fabrication process. Cryogenically compatible MEMS in quantum PICs provide seamless integration of displacement and strain actuators on-chip. This has been shown to be a fundamental part of other PIC building blocks, such as tunable filters[45] and phase shifters[46,47]. Additionally, MEMS have demonstrated compatibility with integrated single-photon emitters for routing[48] and filtering[49,50], and the strain distribution created by capacitive MEMS has been used to improve the spectral overlap of quantum emitters[51,52], and the spin coherence of color centers[53]. The presented platform can be leveraged for these applications, as shown in Fig. 4, in which the quantum emitters are strain-tuned into spectral alignment using a MEMS cantilever.

Applications of the presented device span many relevant quantum optics experiments. The analog tunability of our device enables fine optimization and stabilization of beam splitting ratios, critical for random number generation and accurate HBT and Hong-Ou-Mandel (HOM) measurements. Fast reconfiguration in the MHz regime can be used to tap and measure properties of quantum emitters on-chip. This can be readily applied to existing quantum communication protocols such as quantum key distribution to identify multiphoton or blinding attacks. Additionally, by changing the device geometry to mismatch the waveguide modes (i.e. reducing the width of one of the waveguides), the device can act as a phase shifter[45], which, together with passive beam splitters, form a complete set

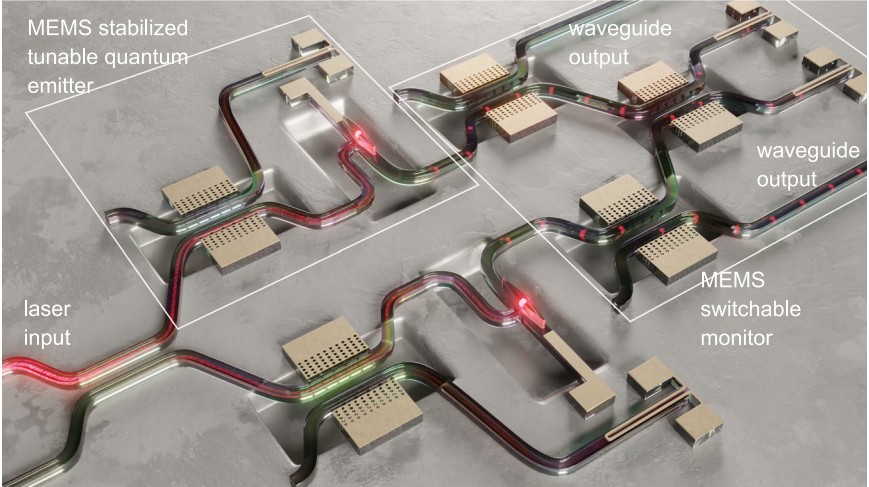

**Fig. 4 Near-term application of our technology.** Example of a monitored and stabilized on-chip photon source as a near-term application of our technology. The device includes two power stabilizers connected to two MEMS-tunable quantum emitters and MEMS splitters for switching into an HBT/HOM monitoring circuit.

of components for arbitrary linear optics and linear optical quantum computation[8]. Figure 4 shows a monitored and stabilized on-chip single-photon source as a near-term application of our technology. The device consists of three parts: stabilized laser excitation of tunable single-photon emitters, switching between monitors and output, and monitoring circuit. A laser input is sent through a passive ~50:50 beam splitter into two $\pi$-pulse power stabilizers each formed by a MEMS splitter and an SNSPD. The laser pulses then excite two MEMS-tunable single-photon sources, and their emission is routed into MEMS beam splitters which act as a switch between the output waveguides and the integrated monitoring. The on-chip monitoring circuit is based on a single MEMS beam splitter terminated with two SNSPDs. This enables switching between the analysis of three crucial aspects of a single-photon source: (i) emitter intensity by routing all the emitted photons into an SNSPD (MEMS splitter in 100:0 splitting), (ii) photon anti-bunching from each emitter (MEMS splitter in 50:50) by routing only the photons from one emitter into the device forming an HBT setup, and (iii) photon indistinguishability (MEMS splitter in 50:50) by routing the photons from both emitters into quantum interference forming a HOM setup. A HOM measurement can then be performed by frequency shifting the spectrum of the emitted single photons of one of the emitters with respect to the other one by means of strain[54]. The device can thus discretely or continuously monitor emitter properties by switching from on-chip to off-chip configurations or by using the MEMS splitters as a tap.

In conclusion, we have demonstrated the on-chip compatibility of MEMS reconfigurable photonics with superconducting single-photon detectors and used it to develop three key elements of quantum optics experiments. We measured routing of classical and quantum light with a high extinction ratio, a photodetector with high dynamic range, and input power stabilization. Using our device, we performed on-chip lifetime and, paired with a fiber-coupled commercial detector, second-order autocorrelation measurements on single photons from a quantum dot source. Our results show that the combination of MEMS and SNSPDs enables the on-chip integration of not only the main building blocks of quantum optics, but also devices for adaptive control, monitoring, and stabilization of classical and quantum optics. The presented technology can overcome current roadblocks towards large-scale quantum optics, and foster applications in quantum communication, metrology, computing, and simulation.

## Methods

**Sample fabrication.** The sample fabrication started with a 250-nm thin film of stoichiometric $Si_3N_4$ on 3.3 μm $SiO_2$ on a silicon handle substrate, provided by a foundry (Rogue Valley Microdevices). We deposited a 9 nm film of $Nb_{0.86}Ti_{0.14}N$ by reactive co-sputtering from separate Nb (200W, DC) and Ti (200W, RF) targets at room temperature in nitrogen and argon atmosphere[21]. After Cr/Au marker lift-off, aligned electron-beam lithography using 350 nm thin ma-N 2403 negative tone photoresist, followed by $CF_4$-based reactive ion etching resulted in patterned NbTiN nanowires, contacts, and MEMS electrodes. A second electron-beam lithography with the same resist and a $CHF_3$-based reactive ion etching yielded the $Si_3N_4$ waveguide devices. Then, we used optical lithography with positive-tone resist to open windows for a wet buffered hydrofluoric acid (BHF) process under-etching the $SiO_2$ film under the MEMS actuators and grating couplers. To avoid the collapse of the suspended actuators due to capillary forces, we dried the sample using critical point drying (CPD). Detailed information of the fabrication, process charts, and additional imaging can be found in the Supplementary Information Section I.

**Measurement setup.** The sample was silver-glued to a custom-designed printed circuit board (PCB), and wirebonded using an Al wedge bonder. The PCB was then mounted in a dilution cryostat (Bluefors) with optical window access and a sample stage temperature below 100 mK. The PCB was connected to coaxial cables leading to room temperature, where the SNSPDs are biased and amplified using a commercial driver system (Single Quantum Atlas). The MEMS components are either directly driven using a high-voltage power supply (Keithley 2410) or through a function generator amplified using a high-speed high-voltage amplifier (Falco Systems, WMA-300). Light was coupled into the chip through an objective (50x, NA 0.82, Partec) inside the cryostat from a CW-laser at 795 nm wavelength, a pulsed laser (795 nm, 2 ps) or the quantum dot single-photon source. The polarization was controlled using a $\lambda/2$ - waveplate on the input path. Detailed information on the individual experiments can be found in the Supplementary Information Sections IV, VII, VIII, X, XI, and XII.

**Optical numerical simulations.** We use a commercial finite element method (FEM)-based Eigenmode solver (COMSOL) to simulate the optical modes for the directional coupler and the absorption within the waveguide section below the SNSPD. The complex refractive index for NbTiN at 795 nm is $n_{NbTiN} = 2.6468$, $k_{NbTiN} = 2.7622$[55], for silicon nitride $n_{Si_3N_4} = 2$ and for silicon dioxide $n_{SiO_2} = 1.45$, both with no imaginary part. More information on our simulation parameters and results can be found in the Supplementary Information Section III.

## Data availability

Source data available at https://doi.org/10.5281/zenodo.4162503.

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

## Acknowledgements

The authors thank Dr. Umer Shah and Prof. Joachim Oberhammer for the amplifier, and Aleksandr Krivovitca for his help with the CPD. This project has received funding from the European Union's Horizon 2020 research and innovation program under grant agreement No. 820423 (S2QUIP). S.G. acknowledges funding from the Swedish Research Council under Grant Agreement No. 2016-06122 (Optical Quantum Sensing). A.R. acknowledges the support of the LIT Secure and Correct Systems Lab, financially supported by the State of Upper Austria, and the Austrian Science Fund (FWF): P29603, I4320. V.Z. acknowledges funding by the Knut and Alice Wallenberg Foundation (KAW, 'Quantum sensors') and the Swedish Research Council (VR, grant No. 638-2013-7152 and grant No. 2018-04251). C.E.-H. acknowledges funding from the Swedish Research Council (2019-00684). K.D.J. is currently with the Department of Physics, Paderborn University, Paderborn, Germany. C.E.-H. is currently with the Research Laboratory of Electronics, Massachusetts Institute of Technology, Cambridge, USA.

## Author contributions

C.E.-H. conceived the device, and C.E.-H., S.G., V.Z., and K.D.J. conceived the demonstrator experiments. C.E.-H. performed the design and simulations. C.E.-H., S.G., J.Z., A.W.E, and S.S. developed and performed device fabrication. S.F.C.S. and A.R. developed and performed the quantum dot growth and characterization. S.G., C.E.-H., L.S., and K.D.J. performed the measurements. C.E.-H. and S.G. wrote the manuscript, and all authors revised the manuscript. C.E.-H., V.Z., and K.D.J. supervised the project.

## Funding

## Competing interests

The authors declare no competing interests.

**Additional information**

