## [Peer Review File · Nature Communications]

Reviewers' Comments:

Reviewer #1:

Remarks to the Author:

Reconfigurable optical chips are promising for various applications in quantum and classical optics. The manuscript "Reconfigurable quantum photonics with on-chip detectors" by Gyer et al demonstrates the first successful integration of a MEMS switch along with superconducting nanowire detectors on a SiN/SiO₂ chip. The authors achieve nearly 30db of extinction in the switching of a reconfigurable beamsplitter on the same chip as the SNSPDs, and show some applications of such an integrated circuit. The manuscript also discusses some prospects for a chip with integrated single-photon sources. Reconfigurable photonics chips for linear optical quantum computing are often based on coupled beam splitters and thermal phase-shifters. In this respect, the authors have taken a different approach by using a MEMS device, which brings along the benefits that are mentioned in the introduction section.

Overall, I find the work interesting and of high importance for applications in quantum optics and linear quantum information processing. The article will likely attract the attention of researchers from other fields such as photonic integrated circuits as well as classical optics. For instance, the authors demonstrate a high dynamic range power meter, or power stabilization, which might be interesting for on-chip sensing. The manuscript is well written and concise, and I would recommend its publication of the article in Nature Communications provided that the authors can address the following concerns satisfactorily:

1- Superconducting nanowires are notoriously hard to fabricate with high yield. So are the MEMS actuators. What is the yield of the fabrication process including the MEMS and the SNSPDs? Can the authors envision a larger device with multiple detectors and actuators? One or two sentences on this topic might be interesting for the readers.

2- In line with question #1, judging from Fig. 1d, the extinction of the beamsplitter on the arm connecting to detector B is not as high as the one connecting to detector A. This is also mentioned in the supplementary. What is the extinction of on the second detector? Please quote it in the main text. Do the authors have an explanation for the seemingly low extinction in this arm? In combination with the low efficiency of detector B, one may lead to the conclusion that the power that is re-routed from detector A is not going to detector B. The authors should clarify this. I suggest that the authors include figure 1d without any normalization in the supplementary.

3- The switching voltages are seriously large, 200Vs! This is the case because of the 3-4 micrometer distance between the beamsplitter arms and the substrate. Is there any particular reason that the authors do not use some of the conventional MEMS designs? Or alternatively, the voltage could be applied between the two arms of the beamsplitter to move the arms closer. Is there any reason to avoid this? What are the prospects for reducing these voltages to a few volts?

4- The manuscript includes a discussion about ac power consumption. This is interesting, but DC power consumption is more important for many applications, for instance linear optical quantum computing. What is the DC power consumption for a fixed configuration of the device, say at 50% split ratio? What about the power consumption at full inversion the signal, that is at 200Vs?

Here are a few more points that may help the authors to improve the article and make it easy for the readers to follow:

1- What is the estimated overall efficiency of the chip? Could the authors separate the losses in different parts of their chip? On a related note, most of the numbers on the plot axes are given in normalized units. In some sections, this makes the plot easier to understand, e.g. Fig. 1b. However, in certain parts, it removes interesting information from the plot, for instance in Fig. 2c. Could the

authors provide the actual input power in nano Watts? This will include some interesting information about the overall efficiency of their chip.

2- Please provide clear numbers for the detection efficiency of each individual detector in the main text. These numbers are in the supplementary, but it is hard to dig them out. For instance, in line 61-62 page 4, "The two detectors feature and different on-chip detection efficiencies". It would be nice if the authors could quote the numbers directly there. Also, I believe these numbers deserve a spot in Fig. 1c as well.

3- On the same line, could the authors quote the extinction for the two arms of the beamsplitters in the main text? It is certainly ok to use the extinction on detector A in the abstract, but it would be nice if the main text included the numbers for both of the arms.

4- 30db extinction is a nice achievement. Could you also plot the data in Fig. 1d in logarithmic scale? It will make the number much easier to appreciate.

5- Again Fig 1d, the data is of high quality and low noise. Using dots or some markers along with a line can make it more clear that these are experimental data points. Otherwise, it may be mistaken for theory curves.

6- Figure 2a, the definition of "amplitude" is unclear in the main text. One has to read the supplementary in detail to understand what is presented there. I suggest that the authors try to clarify the measurements in the main text.

7- The authors present really interesting applications for their chip. I personally find all of the presented results interesting and exciting. However, all these demonstrations have made the paper crowded. I would suggest that the authors move some of these applications to the supplementary and only refer to the results in the main text. For instance, they could move the part with power stabilization to the supplementary.

8- The word "Sensible" on page 7 line 120 seems like a typo. I believe the authors meant "sensitive".

9- The authors measure the lifetime of the single photons using on-chip detectors and quote the lifetime with a precision of 2ps. However, these detectors have a jitter around 121ps. Could the authors elaborate on this?

10- How does the actuator voltage effect the dark counts on the detectors? Do the authors observe a correlation between a DC actuation voltage and the dark counts on the detectors?

Reviewer #2:

Remarks to the Author:

Manuscript by Gyger and coworkers demonstrates photonic circuits with SNSPDs together with mechanical elements to program the circuit. This is an important goal for on-chip quantum optics since for the envisioned large-scale circuits such as quantum computers, all elements have to be integrated on the same chip. Currently, heaters are typically used to change the circuits which are intrinsically incompatible with the low temperatures required for the superconducting devices. Hence proof-of-principle device by Gyger et al. is an important step forward for the rapid developments in the field of on-chip quantum optics. Perhaps the paper does not answer any big scientific questions, but the demonstration itself should appeal to a wide range of readers from fields of quantum optics and photonics. However, there is still a number of major improvements that have to be implemented before I can recommend the publication.

The authors often use the normalized units, without explaining the normalization. For example, are the values in Fig. 1C and 1D the same, or different? (and what was the bias current there?) Why does the DCR of det. A show a sudden drop near a normalized bias current of 1? This suggests transition to normal state below normalized bias current of 1. Are the counts of det. B at low voltage in 1d the DCR or "real" clicks. When taking the efficiencies $\eta_{A,B}$ into account, does the total incoming photon rate $R_A/\eta_A + R_B/\eta_B$ stay constant or does it change with V? In other words, is there an increase in insertion loss with voltage? Same for Fig. S7. Here $8e5$ crosstalk counts sounds bad, but in a.u. there is no way to compare it to the actual rate, to the DCR, or even between the two detectors. Please also comment on the shape – and its origin – of frequency dependence of the unbiased SNSPD readout.

At several points, I found that the text was lacking details on how certain numbers were obtained, e.g. as power dissipation. It is only at the end, or sometimes not at all, that the authors refer to the SI. Please provide more references to the SI and/or do this earlier in the discussions. Staying with the point of the dissipation, the authors need to discuss in more detail where this happens. Device is capacitor which only stores energy. Dissipation may occur in the (superconducting or normal?) electrodes on the chip, but most likely it happens inside the voltage source and is therefore largely irrelevant for the device operation at low temperature.

Typos: half missing in Ec. $P_{\text{leak}} = V^2/R$

In Fig. 2C, please indicate in the caption the meaning of different things visible in the cartoons.

In Fig. S3, the curves are inverted compared to the main text. Is this real, or only a mistake in the colors used? Why is there coupling ratio 100/0 (or 0/100) at $V=0$? Is this carefully designed, accidental, or a starting assumption of the model. Also, as far as I remember Si₃N₄ is high stress material so why is it assumption no stress?

What kind of e-beam resist is used for the SNSPDs? Please add this to the Sec. 2 of the SI. For the detector operation, it is mentioned that the HF does not etch the NbTiN, but this would remove e.g. HSQ from the detectors. Does this affect detectors in any way? Is there e.g. a risk of oxidation without resist?

Typically, the pulses from the SNSPD are analyzed with TCSPC electronics to find e.g. the count rate. However, for PID feedback in Fig. 2B, one needs analog signal. Same for the "Amplitude" in Fig. 2A. (also here, what are the units?). Please explain in text how exactly is this determined: using QuTag, or was different equipment used?

Another point of attention are the symbols in the schematics. Why is there an RF source in Fig. 2B? In Fig. 2C a circle with + and – is used for the voltage source, but in other figures it is a circle with a straight line. Please use a consistent symbol. Also, in the latter, the arrow is somewhat confusing as this is typically reserved for current sources (although one can sometimes see it for V source too).

Although it is explained in the text, I found Fig. 3 slightly misleading as the, very nice, device rendering does not correspond to the experiment shown in other panels. Please provide schematics of the experiments 3a and 3b and place the "outlook" rendering in a separate figure. It is also mentioned to do HOM with this device, but how will the required tunable delay (typically cms) be generated on the chip?

In conclusion, authors have done very nice demonstration but need more clarification.

Reviewer #3:

Remarks to the Author:

The manuscript reports on the integration on the same chip of a low-power MEMS with

superconducting nanowire single-photon detectors (SNSPD). This configuration could be of benefit to the development of future integrated quantum circuits because the MEMS actuators do not generate (or require) heat and are therefore compatible with the heat-sensitive SNSPDs.

The MEMS tunable couplers and SNSPDs are not novel components on their own but their combination on the same chip has not been reported before and therefore contains potential novelty. In particular, the use of the same NbTiN layer to build the actuators the SNSPD, the electrical connections and contact pads is an interesting solution that could find applications in the fabrication of quantum circuits. However, the manuscript does not contain sufficient novelty and impact to meet the standards of Nature Communications for the reasons detailed below.

Figure 3c is just a 'future vision', far from what is demonstrated and discussed in the paper. For example, not all the components are tested together in the same cryostat (the QD emitter is at 5 K, the SNSPDs are at 100mK in a dilution fridge); the Hanbury Brown and Twiss (HBT) measurement is performed by routing the photons from the source into an off-chip beam splitter; only one arm is coupled into the on-chip detector, while the other arm is coupled into a commercial off-chip detector. As such the paper mostly demonstrates the potential to integrate MEMS and SNSPD on the same chip, but the on-chip measurements are only performed on the individual components.

Another weakness of the paper is that the authors have not succeeded in doing the HBT 2nd order correlation measurement from the QD source with a pair of waveguide integrated SNSPDs. In Fig 3b the black upper trace is with a pair of commercial fibre coupled SNSPDs; the lower blue trace is with one fibre-coupled SNSPD and one waveguide integrated SNSPD. The authors don't give absolute correlation counts or acquisition times, but the lower trace is very noisy and is clearly barely working. So the claim in the conclusion that they did full second order correlation measurements with the waveguide integrated SNSPDs is not true (the abstract doesn't make this claim

From a device point of view there are components that need to be improved or optimised to make this device useful for quantum experiments. For example, the grating couplers have very high losses, which lead to a large discrepancy in the $g(2)$ measurements. Also, the detectors exhibit a very large difference in their efficiencies. These figures are far worse than the performance reported in the literature for similar devices. For example, the authors used grating couplers based on the design Zhu et al., which exhibit far lower losses of $<3\text{dB}$

We thank the reviewers for the time they have spent reading and evaluating our manuscript. This document describes the answers to each of the reviewer's comments. We marked the comments by the reviewers in blue, and highlighted the parts where text was changed in the manuscript in red.

Reviewer #1:

Overall, I find the work interesting and of high importance for applications in quantum optics and linear quantum information processing. The article will likely attract the attention of researchers from other fields such as photonic integrated circuits as well as classical optics. For instance, the authors demonstrate a high dynamic range power meter, or power stabilization, which might be interesting for on-chip sensing. The manuscript is well written and concise, and I would recommend its publication of the article in Nature Communications provided that the authors can address the following concerns satisfactorily:

We thank Reviewer #1 for the appreciation of our work and recommendation of publication in Nature communications. The constructive comments by the reviewer were used to improve our work as described below.

1- Superconducting nanowires are notoriously hard to fabricate with high yield. So are the MEMS actuators. What is the yield of the fabrication process including the MEMS and the SNSPDs? Can the authors envision a larger device with multiple detectors and actuators? One or two sentences on this topic might be interesting for the readers.

We agree with the reviewer that this is a very relevant concern. To make this clearer to the reader, we have included in the text a discussion on our MEMS and SNSPD yields and their yield in the literature.

On the yield of our SNSPD fabrication, we included the following sentence in the Supplementary Section 2: "Previous to our MEMS measurements, we characterized the SNSPDs in our sample in a closed-cycle cryostat at 2.5K, which yielded saturation of internal quantum efficiency at 850 nm for 3 out of 5 tested devices." We also included Fig. S4 to the Supplementary Information, to give the reader data to judge the behavior.

Fig. S4. Photon count rate normalized to the same range used during flood illumination at 850 nm. We pre-characterized the devices after the release step (Figure S3e) at 2.5 K, and observed saturated internal quantum efficiency for 3 out of 5 devices (orange, purple, and red in the plot).

In the literature, higher SNSPD yields have been reported, and we adapted the text in Discussion, paragraph 1 and referenced the extended Supplementary: “While the number of devices in our proof of concept experiment is small to estimate a yield (see Supplementary Section 2), in an industrial fabrication process, fabrication yield is expected to be no limiting factor [42, 43].”

On the yield of our MEMS fabrication, we included the following in Results, subsection 2, paragraph 3: “In addition, the robustness of this geometry in terms of yield is confirmed by the operation of three copies of the same MEMS actuator on the same chip.” Additionally, we included a room temperature measurement of another MEMS splitter on the same chip, in Supplementary Fig. S9b (plotted below).

In the literature, MEMS have been used as the main building block in the largest silicon photonic circuits to date (see Ref. [45] in the main text: *Tae Joon Seok, Kyungmok Kwon, Johannes Henriksson, Jianheng Luo, and Ming C. Wu, "Wafer-scale silicon photonic switches beyond die size limit," Optica 6, 490-494 (2019)*). We added text in our Discussion, paragraph 2 to clearly refer to this work: “For example, capacitive MEMS in silicon have demonstrated sub-10 V actuation [16] and excellent scalability [44], as evidenced by the demonstration of a 240x240 photonic MEMS switch matrix, i.e. the largest silicon photonic circuit demonstrated to date [45].”

We ourselves can envision larger scale devices through industrial-scale clean-room processes and the different requirements on detectors based on their use-case. While the SSPDs for true single-photon correlation measurements need to show high efficiency, the ones used for power stabilization and for power measurements are not so critical. On the MEMS side, photonic MEMS scaling to tens of thousands of devices is possible (> 57000 in [45]), and any non-unity yield can be addressed by redundant PIC architectures so that signals going through a faulty device can be rerouted into a non-faulty one.

2- In line with question #1, judging from Fig. 1d, the extinction of the beamsplitter on the arm connecting to detector B is not as high as the one connecting to detector A. This is also mentioned in the supplementary. What is the extinction of on the second detector? Please quote it in the main text. Do the authors have an explanation for the seemingly low extinction in this arm? In combination with the low efficiency of detector B, one may lead to the conclusion that the power that is re-routed from detector A is not going to detector B. The authors should clarify this. I suggest that the authors include figure 1d without any normalization in the supplementary.

We agree with the reviewer that the difference in extinction is a concern. To address this, we clarified our measurements in the main text and included further analysis and measurements in the Supplementary.

We included the extinction value in the main text under Results, subsection 2, paragraph 2: “[...] yielding a high extinction ratio between detectors of 28.1 dB, and on-off ratio for individual ports of 27.5 dB (detector A) and 11.7 dB (detector B).”

To investigate the reason for the extinction mismatch, we closely imaged our detectors under a high magnification in an SEM. The new SEM images, which we include in the Supplementary in Fig. S2, show a clear constriction in Detector B, which leads to early saturation of the detector due to lower current densities, leading to a much lower detection efficiency. We included a sentence on this in the main text: “The detectors show different on-chip detection efficiencies (Detector A is 44.6 times more efficient than Detector B) [25], which we attribute to a lithographic defect in detector B, which results in a nanowire constriction and lower current densities, and therefore lower detection efficiency in the remaining nanowire (see Fig. S2).”

Figure S2: SEM images of the SNSPD hairpins. It is visible that Device B has a defect in the bend leading to a low critical current, limiting the detection efficiency of the straight section.

To further prove that the MEMS splitter is indeed balanced, we performed analysis of the measurements in Fig. 1d and measured a device consisting of a grating-coupled MEMS splitter at room temperature. We report the results in Fig. S9, and included the following description: “To investigate the cause of the difference in efficiency in our two detectors, we investigate the power balance of our design. Figure S9a shows the sum of the normalized tuning curves in our device. The sum being within 5% of unity indicates that power is conserved during actuation, and thus MEMS tuning does not induce the large difference between SNSPD efficiencies of 44.6 times. To further prove this, and to demonstrate that the efficiency mismatch is not intrinsic to our MEMS splitter, we performed room temperature characterization of a similar MEMS splitter on the same chip that features grating-coupled input and output ports. The results shown in Fig. S9b demonstrate symmetric tuning without count normalization. The lower extinction observed in this device is attributed to the difference in operation temperature, which results in a change in device geometry via relative thermal expansion of the materials forming the cantilever.”

Fig. S9. a) Sum of normalized counts of the device in Fig. 1d show an actuation-independent power conservation within 5%. b) Room temperature measurements of a grating-coupled MEMS splitter on the same chip show symmetric operation with similar counts, further demonstrating that the difference in measured optical power can be attributed to the SNSPDs and not to the MEMS splitter.

We also included the MEMS tuning curves (Fig. 1d) without normalization in Fig. S8.

Fig. S8. Counts in Detector A and B under MEMS actuation.

3- The switching voltages are seriously large, 200Vs! This is the case because of the 3-4 micrometer distance between the beamsplitter arms and the substrate. Is there any particular reason that the authors do not use some of the conventional MEMS designs? Or alternatively, the voltage could be applied between the two arms of the beamsplitter to move the arms closer. Is there any reason to avoid this? What are the prospects for reducing these voltages to a few volts?

We agree with the Reviewer that 200 V is above the usual driving voltage for MEMS devices and the value needs to be reduced for future large scale integration. We clarified the reasons for this high voltage in the main text Discussion, paragraph 2, as follows: “The high voltage actuation required by our device is due to 1) the large capacitive gap between device layer and substrate, and 2) the out-of-plane bending of the device caused by stress relaxation on the Si₃N₄ and NbTiN layers, which requires stiff cantilevers to maintain evanescent coupling between the directional coupler waveguides. We believe that this is not a fundamental limit in our platform, the actuation voltage can be reduced by tuning the stack stress distribution, by designing strain-tolerant MEMS actuators, and by using in-plane actuators with smaller capacitive gaps [16]. For example, capacitive MEMS in silicon have demonstrated sub-10 V actuation [16].”

4- The manuscript includes a discussion about ac power consumption. This is interesting, but DC power consumption is more important for many applications, for instance linear optical quantum computing. What is the DC power consumption for a fixed configuration of the device, say at 50% split ratio? What about the power consumption at full inversion the signal, that is at 200Vs?

DC power consumption is indeed more relevant for most applications. The reason we chose to focus on AC power consumption is that our device is essentially a vacuum-filled capacitor, which consumes very small power under DC bias. To reflect this low power consumption and expand on its description, we added additional calculations and text in Results, subsection 2, paragraph 3: “Under DC actuation, the power consumption is driven by leakage currents. Due to the high insulation of vacuum and SiO₂ in our capacitor, leakage currents are minimal, which leads to power dissipation in the femtoWatt-range (see Supplementary Section 5) despite the high voltages applied. For example, for a 50:50 beam splitting ratio we estimate the power dissipation to be 6 fW, and for full inversion of the signal, 8.5 fW.” We also included a sentence in the Supplementary Information Section 10 on the prospects for scaling up this system in terms of power consumption: “In a large-scale PIC based on these actuators, however, the

power consumption will be most likely dominated by the control electronics and the routing of electrical lines.”

1- What is the estimated overall efficiency of the chip? Could the authors separate the losses in different parts of their chip? On a related note, most of the numbers on the plot axes are given in normalized units. In some sections, this makes the plot easier to understand, e.g. Fig. 1b. However, in certain parts, it removes interesting information from the plot, for instance in Fig. 2c. Could the authors provide the actual input power in nano Watts? This will include some interesting information about the overall efficiency of their chip.

We agree with the referee that adding overall efficiency values adds to the understanding of the reader. We estimated our overall loss from before the cryostat window to detector A to be 33.7 dB from the power meter measurements. We added this information to the Supplementary section 5 with the following text: “From the power meter measurements, we calculate a system detection efficiency from the cryostat window to the detector of -33.7dB for detector A and -49.3 dB for detector B, when all the power is routed to the respective detectors.”

For separating the losses we can only give estimates. We characterized a loop-back test device from the same chip with grating coupler inputs and outputs, yielding 20 dB loss. This value is largely dependent on fine alignment between the input beam and the grating couplers, and thus it should be taken as a rough estimate. Due to our largely mode-mismatched input, we believe most of the loss comes from the input coupling, as described in Zhu et al (Ref. [1] in the Supplementary). Assuming the output coupler features 3 dB loss as in Zhu, and assuming perfect detection efficiency in Detector A, our device routing shows 17 dB loss. We believe this high loss is dominated by scattering at the unoptimized waveguide anchors in the splitter and/or by NbTiN residues from a non-optimal lithography. Anchor loss is a common problem with fully suspended waveguide devices, and can be solved by co-optimization of design and fabrication or by membrane clamping (see discussion on sources of scattering loss in photonic MEMS devices in ref. [16]: *C. Errando-Herranz, A. Y. Takabayashi, P. Edinger, H. Sattari, K. B. Gylfason and N. Quack, "MEMS for Photonic Integrated Circuits," in IEEE JSTQE, vol. 26, no. 2, pp. 1-16, Art no. 8200916, 2020*). We plan to reduce loss following these approaches in future fabrication runs. We would like to highlight that in this work we focus on proof of principle demonstrations, and high on-chip coupling efficiencies are not within the expertise of our group and not prioritized in the construction of our setups. We think this should not distract from the results. As the reviewer requested, we added the input power that was used for normalization to Fig. 2c (3.15uW).

To highlight the theoretical absorption efficiency of our SNSPDs, we performed optical simulations yielding an SNSPD absorption efficiency of 95.5%, which indicates high detection efficiency given the near-unity internal efficiency indicated by the PCR plateau. To describe this, we included a new section in the Supplementary with the following text: “We performed simulations of waveguide-coupled SNSPD with an eigenmode solver (COMSOL Multiphysics) using the NbTiN material parameters in Banerjee et al. [4]. For our fabricated waveguide cross-section, the simulated absorption is -0.675 dB/ μm . Our SNSPDs are 20 μm long, yielding an absorption of -13.5 dB, or 95.5 %.”

2- Please provide clear numbers for the detection efficiency of each individual detector in the main text. These numbers are in the supplementary, but it is hard to dig them out. For instance, in line 61-62 page 4, “The two detectors feature ... and different on-chip detection efficiencies”. It would be nice if the authors could quote the numbers directly there. Also, I believe these numbers deserve a spot in Fig. 1c as well.

We included the efficiency difference between the detectors in the main text: “The detectors show different on-chip detection efficiencies (Detector A is 44.6 times more efficient than Detector B)” and in the caption in Fig. 1. Additionally, as mentioned above, we performed optical simulations yielding a SNSPD absorption efficiency of 95.5%, and we performed SEM imaging of the detectors and found out about a constriction in Detector B, likely the cause for its 44.6 times lower efficiency (new Figure S2).

3- On the same line, could the authors quote the extinction for the two arms of the beamsplitters in the main text?

We extended the paragraph to contain the values: “The measured tuning curve is shown in Fig. 1d, and follows our simulations (see Supplementary Fig. S8), yielding a high extinction ratio between detectors of 28.1 dB, and on–off ratio for individual ports of 27.5 dB (detector A) and 11.7 dB (detector B).”

4- 30db extinction is a nice achievement. Could you also plot the data in Fig. 1d in logarithmic scale? It will make the number much easier to appreciate.

We thank the reviewer for recognizing the high extinction ratio. To highlight this result, we split the plot in Fig. 1d and added the extinction ratio between the two arms in logarithmic scale and adapted the Figure caption to reflect this change.

“Measured photon detection counts versus MEMS actuation voltage, normalized to the individual maximum transmission and extinction ratio (ER) between detectors.”

5- Again Fig 1d, the data is of high quality and low noise. Using dots or some markers along with a line can make it more clear that these are experimental data points. Otherwise, it may be mistaken for theory curves.

We appreciate the reviewer recognizing the data quality and added dots to make clear that these are measurement values.

6- Figure 2a, the definition of “amplitude” is unclear in the main text. One has to read the supplementary in detail to understand what is presented there. I suggest that the authors try to clarify the measurements in the main text.

We thank the reviewer for pointing out this potentially confusing description. We extended the main text as follows and also highlighted the SI for the reader: “The normalized frequency response of our device is shown in Fig.2a, where the amplitude $|A|(\omega)$ is defined as the normalized amplitude ($|A|(\omega) / |A|(\omega \rightarrow 0) = 1$) of the modulated optical transmission under a sinusoidal actuation voltage with amplitude $\Delta V(\omega)$ (see schematic in the inset of Fig. 2a, and further description in the Supplementary Section 8).”

7- The authors present really interesting applications for their chip. I personally find all of the presented results interesting and exciting. However, all these demonstrations have made the paper crowded. I would suggest that the authors move some of these applications to the supplementary and only refer to the results in the main text. For instance, they could move the part with power stabilization to the supplementary.

We are very happy that the reviewer likes our application demonstrations. While we do not want a too crowded paper, we kindly suggest that the direct feedback from on-chip measurements to on-chip manipulation is an important demonstration that we would prefer in the main text. However, if the referee feels strongly about that issue, we are open to move the demonstration.

8- The word “Sensible” on page 7 line 120 seems like a typo. I believe the authors meant “sensitive”.

We fixed the mistake. Thank you.

9- The authors measure the lifetime of the single photons using on-chip detectors and quote the lifetime with a precision of 2ps. However, these detectors have a jitter around 121ps. Could the authors elaborate on this?

We thank the referee for highlighting this possible point of confusion. In our measurement we fit the lifetime measurement to a lifetime of a transition convoluted with the Instrument Response Function (the jitter) of our SNSPDs. The IRF is then part of the fitting routine and describes the jitter behavior of the device. We extended the manuscript by adding the information on the IRF: “We extract the lifetime by fitting a decay convoluted with the Instrument Response Function of the detector measured at the same wavelength using a 3 ps pulsed laser. (See Supplementary Section 5)”.

And described our fitting model in the Supplementary: “We extract the lifetime by fitting the fit function (Eq. 5) including the Instrument Response Function (IRF) (see Fig. S6) of the detector.

$$f(t) = A \cdot \text{IRF}(t) * \left(H(t) \cdot \exp\left(\frac{-t}{\tau_{XX}}\right) * H(t - t_0) \cdot \exp\left(\frac{-(t - t_0)}{\tau_X}\right) \right)$$

τ_x is the Exciton lifetime, H is the Heaviside function and * denotes the linear convolution. Due to the X being fed by the XX decay, an additional decay with a fixed lifetime τ_{xx} extracted from supporting measurements is part of the fitting function.”

The uncertainty of the parameter we give in the manuscript is estimated from the quality of the fit parameters and to make this clear to the reader we added the following sentence to the Supplementary Information: “The fit is made using LMFIT [8] and the uncertainty given for the parameters is estimated from the fit quality.”

10- How does the actuator voltage effect the dark counts on the detectors? Do the authors observe a correlation between a DC actuation voltage and the dark counts on the detectors?

We observed no influence of DC actuation voltage on the dark counts of the detectors. To clarify this to the reader, we added measurement data in the Supplementary as Fig S12, where we show the DCR for a pre-characterized SNSPDs under different actuation voltages. We added the following sentence to the SI: “We investigated the behavior for DC actuation and we observe no additional dark counts generated by the biasing of the MEMS actuator (Figure S12).”

Fig. S12. Dark counts under 0 V, 80 V and 110 V static actuation. This data was measured in a pre-characterization experiment and no change in dark counts was observed.

For AC actuation voltages we highlighted the problem for unbiased detectors in the Supplementary Information, Fig S11. The measurement data contained no counts below 10 kHz, which is why we reduced the plotted regions to the one shown in the SI.

We thank Reviewer #1 for the constructive comments and hope we answered them satisfactorily.

Reviewer #2:

This is important goal for on-chip quantum optics since for the envisioned large-scale circuits such as quantum computers, all elements have to be integrated on the same chip. Currently, heaters are typically used to change the circuits which are intrinsically incompatible with the low temperatures required for the superconducting devices. Hence proof-of-principle device by Gyger et al. is an important step forward for the rapid developments in field on chip quantum optics.

We thank Reviewer #2 for seeing the value in the proof-of principle device we show in this publication and for the constructive criticism. We addressed the questions below.

The authors often use the normalized units, without explaining the normalization. For example, are the values in Fig. 1C and 1D the same, or different? (and what was the bias current there?)

We agree with reviewer #2 that normalized units should be explained and we now do this in the answer and the Manuscript accordingly, as explained below.

Fig 1c and 1d are differently normalized, due to the different points we want to make in this figure. Fig 1c is normalized to the saturation of the internal detection efficiency of the individual detectors. We want to show in this subfigure that the detectors even after the BHF release step show high intrinsic detection

efficiency. Due to the difference in on-chip detection efficiency of the two devices, we normalize them to their individual values.

We added a note in the figure caption to make this clear: “Photon count rate at a wavelength of 795 nm for the two on-chip detectors and the corresponding dark counts, **normalized to their individual saturated detection.**”

In Fig 1d we want to show the extinction of the beam splitter, and therefore we normalized the figure to the maximal transmission for each individual arm. To make this clear to the reader, we changed the figure caption to read: **d) Measured photon detection counts versus MEMS actuation voltage, normalized to the individual maximum transmission and extinction ratio (ER) between detectors.**

To give the reader information of the measurement point, we added the biasing for the jitter measurements of the detectors to the Supplementary: **The detectors were biased at 0.8 I_c with 36 dark counts per second (detector A) and 0.88 I_c with 25 dark counts per second (detector B).**

This is the same biasing for the MEMS measurements, which we noted in the Supplementary with: **The detectors were biased as in the jitter measurement described above.**

Why does the DCR of det. A show a sudden drop near a normalized bias current of 1? This suggests transition to normal state below normalized bias current of 1.

The normalization of the axis is necessary in this Fig. 1c, due to the large difference in critical current of 15.8 μA (detector A) and 5.9 μA (detector B). Therefore we chose a critical current value to normalize to, in our case the PCR curve. Subsequent measurements of critical currents do not lead to the exact same result. This is characterized e.g. in Fig. 5 of Mattioli et al. (Ref. [24] in the main text: *Mattioli, F., R. Leoni, A. Gaggero, M. G. Castellano, P. Carelli, F. Marsili, and A. Fiore. “Electrical Characterization of Superconducting Single-Photon Detectors.” Journal of Applied Physics 101, no. 5 (March 1, 2007): 054302. <https://doi.org/10.1063/1.2709527>). In our case this spread is 0.2 μA at detector A, well within the expected range. We extended Fig. 1 caption, to explain this fact: **“The reduced critical current in the dark count measurement of detector A is due to measurement-to-measurement fluctuations in the critical current.”***

And extended the manuscript by adding one sentence and a reference to the mentioned paper: **“The difference in critical current between dark and illuminated measurement is linked to the stochastic nature of the superconducting to normal state transition, leading to a spread in the measured switching currents [24].”**

Are the counts of det. B at low voltage in 1d the DCR or “real” clicks.

The counts on detector B are real clicks, and we clarified this in the Supplementary: **“For the MEMS characterization in Fig. 1d, our absolute measured counts ranged from 4 400 246 to 7 793 for detector A, and 98 619 to 6 747 for detector B (4 324 665 and 7 439 respectively at zero actuation voltage), well above the DCR”.**

When taking the efficiencies $\eta_{A,B}$ into account, does the total incoming photon rate $R_A/\eta_A + R_B/\eta_B$ stay constant or does it change with V ? In other words, is there an increase in insertion loss with voltage?

We thank Reviewer #2 for this important comment. To investigate the reason for the extinction mismatch, we imaged our detectors under a high magnification in an SEM, further analyzed our data, and performed additional measurements in a test device. The new SEM images, now Supplementary in Fig. S2 (see

above in the reply to Reviewer #1), show a clear constriction in Detector B, which leads to early saturation of the detector due to lower current densities, leading to a much lower detection efficiency. We included a sentence on this in the main text: “The detectors show different on-chip detection efficiencies (Detector A is 44.6 times more efficient than Detector B) [25], which we attribute to a lithographic error in detector B, which results in a nanowire constriction and lower current densities, and therefore lower detection efficiency in the remaining nanowire (see Fig. S2).”

Moreover, we further analyzed our data with this aspect in mind, and show the results in Fig. S9a, which are stable within 5%. We also characterized a grating-coupled MEMS splitter on the same chip (Fig. S9b, shown above) at room temperature, yielding symmetric tuning, which points towards the lower efficiency being caused by the detectors rather than by the MEMS splitter. We added a section to the Supplementary, with part of the text shown below: “To investigate the cause of the difference in efficiency in our two detectors, we investigate the power balance of our design. Figure S9a shows the sum of the normalized tuning curves in our device. The sum being within 5% of unity indicates that power is conserved during actuation, and thus MEMS tuning does not induce the large difference between SNSPD efficiencies of 44.6 times. To further prove this, and to demonstrate that the efficiency mismatch is not intrinsic to our MEMS splitter, we performed room temperature characterization of a similar MEMS splitter on the same chip that features grating-coupled input and output ports. The results shown in Fig. S9b demonstrate symmetric tuning without count normalization. The lower extinction observed in this device is attributed to the difference in operation temperature, which results in a change in device geometry via relative thermal expansion of the materials forming the cantilever.”

Same for Fig. S7. Here $8e5$ crosstalk counts sounds bad, but in a.u. there is no way to compare it to the actual rate, to the DCR, or even between the two detectors. Please also comment on the shape – and its origin - of frequency dependence of the unbiased SNSPD readout.

To make it possible for the reader to judge the different signal levels, we included the PCR normalization in the Supplementary section 5: “The absolute detector counts in Fig. 1c are 190 472 (Detector A), and 45 955 (Detector B) for a DCR of 100 counts.”

We unfortunately can not give a satisfying answer on the shape of the frequency dependence, as we do not know the origin beyond the fact that we observe cross talk on the leads. The leads are shielded coaxial cables with a shared common ground for SNSPDs and the MEMS actuation. The counts are not created by signal-shapes like seen in Supplementary Figure S5, but crosstalk that triggers an event on our time-to-digital converter. The exact point linking the different leads to create the cross-talk is unclear.

At several points, I found that the text was lacking details on how certain numbers were obtained, e.g. as power dissipation. It is only at the end, or sometimes not at all, that the authors refer to the SI. Please provide more references to the SI and/or do this earlier in the discussions. Staying with the point of the dissipation, the authors need to discuss in more detail where this happens. Device is capacitor which only stores energy. Dissipation may occur in the (superconducting or normal?) electrodes on the chip, but most likely it happens inside the voltage source and is therefore largely irrelevant for the device operation at low temperature.

We have now added more references to the Supplementary earlier in the individual discussions.

We thank the reviewer for the suggestion on power consumption location. Indeed, the resistance along our lines is very low (most metals are superconductive, and the non-superconductive parts, largely the PCB, have a very low resistance below 0.1Ω), but the source features a $10 \text{ M}\Omega$ resistance. To highlight this point, we included a sentence in the main text: “This power, only consumed during dynamic actuation, is dissipated along the non-superconducting transmission line, which is limited to the off-chip components and in particular the high-resistivity voltage source, and thus far from our SNSPDs.” and in Supplementary section 10: “Our circuit comprises the capacitance of our actuator followed by series resistors from the electrical wiring and connections, and the voltage source. Since the on-chip wirebonds are made by superconducting aluminum and the PCB features a resistance below 0.1Ω , the heat dissipation likely occurs in the high-resistivity voltage source ($10 \text{ M}\Omega$).”

Typos: half missing in Ec. $P_{\text{leak}} = V^2/R$

We thank the reviewer for finding this error, it is now corrected in the SI.

In Fig. 2C, please indicate in the caption the meaning of different things visible in the cartoons.

We added the following explanation to the caption: “The insets show the active detectors and MEMS settings in each of the three ranges: for lowest input power, Detector A is used with most of the power routed into it. For medium input powers the MEMS splitter routes most of the optical power into lower-efficiency Detector B, and both detectors can be used. For highest optical power, no actuation is applied and low-efficiency Detector B is used.”

In Fig. S3, the curves are inverted compared to the main text. Is this real, or only a mistake in the colors used? Why is there coupling ratio 100/0 (or 0/100) at $V=0$? Is this carefully designed, accidental, or a starting assumption of the model.

The colors were inverted and now are fixed, we thank the reviewer for finding this mistake. The directional coupler in this work was designed to cover a number of mode interference periods to ensure efficient coupling ratio tuning, and so the reason why the splitting ratio is largely unbalanced at $V=0$ is due to the length matching a multiple of the mode interference period. The coupling ratio in the MEMS simulation is thus set by geometry (waveguide cross-section and coupling length), but it was not intentionally designed, although it is possible to design it accurately as any other directional coupler. This is evidenced by the fact that our simulation approximates reasonably well our experimental results. We clarified this in the Supplementary section 6, paragraph 7: “Note that the initial coupling ratio in our simulation is set by our fabricated geometrical parameters, and the simulation assumes a constant directional coupler waveguide cross-section along the propagation direction for each actuator voltage.”

Also, as far as I remember Si₃N₄ is high stress material so why is it assumption no stress?

The reviewer is correct that Si₃N₄ is usually a high stress material. In our geometry the cantilever is clamped on one side, which leads to relaxation of the film stress through strain i.e. by contracting or expanding along the cantilever length. This makes the assumption of no-initial stress valid. To make this clear to the reader we added the following sentence to Section 6 of the Supplementary: “The assumption of no initial stress holds for our system, since, although Si₃N₄ (and in our case, NbTiN) often presents significant internal tensile stress, single-clamped suspended structures like the actuators fabricated in our

work relax film stress through strain (i.e. by contracting or expanding along the cantilever length), which drastically reduces their internal stress.”

What kind of e-beam resist is used for the SNSPDs? Please add this to the Sec. 2 of the SI. For the detector operation, it is mentioned that the HF does not etch the NbTiN, but this would remove e.g. HSQ from the detectors. Does this affect detectors in any way? Is there e.g. a risk of oxidation without resist?

We used ma-N2403 from microresist technology as e-beam resist. We expanded the sentence in the Supplementary as follows: “After Cr/Au marker lift-off (Fig. S3b), aligned electron beam lithography using 350nm thin ma-N 2403 negative tone photoresist, followed by CF4-based reactive ion etching resulted in patterned NbTiN nanowires, contacts, and MEMS electrodes (Fig. S3c).”

Our SNSPDs are protected from the BHF step by resist, and thus we do not observe degradation. For the exposed MEMS electrodes, we observed that BHF does not etch NbTiN, but removes the oxide on the metal surface. We rely on the self-limiting oxidation of NbTiN films (see Ref. 3 in the Supplementary: Zhang, Lu, Lixing You, Liliang Ying, Wei Peng, and Zhen Wang. “Characterization of Surface Oxidation Layers on Ultrathin NbTiN Films.” *Physica C: Superconductivity and Its Applications* 545, no. Supplement C (February 15, 2018): 1–4. <https://doi.org/10.1016/j.physc.2017.10.008>.), that together with the high initial thickness of 9 nm allows for stable electrodes. We added information to the Supplementary to make this fact clear to the reader: “We expect the BHF process to etch the thin native oxide film on the surface of the NbTiN, self-limited to 1.3 nm [3]. This oxide film would grow again after exposure to atmospheric oxygen, resulting in a slightly thinner superconducting film [3]. The reduction in thickness, well below a nanometer, might result in a slight degradation of the superconducting properties of the exposed areas of our 9 nm thick NbTiN film [3]. However, in our fabrication process, our SNSPDs were protected by resist, and thus we observed no degradation of their properties.”

Typically, the pulses from the SNSPD are analyzed with TCSPC electronics to find e.g. the count rate. However, for PID feedback in Fig. 2B, one needs analog signal. Same for the “Amplitude” in Fig. 2A. (also here, what are the units?). Please explain in text how exactly is this determined: using QuTag, or was different equipment used?

We thank the reviewer for pointing out this missing information. For the measurement in Fig. 2b we use the count rate data we get from the driving electronics fabricated by SingleQuantum with a time binning of 0.1 s. We then use this measured value as input to the PID loop implemented in LabView that controls a DA converter to stabilize this value to a chosen goal value (in our case 150 000 cts/s). To make this clear to the reader, we added the following sentence to the caption: “The detection events are counted by the driving electronics of the detectors.”

and to the main text: “A PID feedback loop uses the detection counts provided by the SNSPD driver electronics to stabilize the measured power without manual intervention.”

The SI section on power stabilization was extended as follows: “The current detection count rate measured by the SNSPD driver was read out every 100 ms (currently limited by the counting electronics) and a PID control loop stabilized the applied voltage with the photon detection events as the process variable.”

Same for the “Amplitude” in Fig. 2A. (also here, what are the units?).

The measurement in Fig. 2a is now explained in depth in the Supplementary section 8. The counts were measured using a qutag correlator from qutools and evaluated using ETA (<https://timetag.github.io>). The

trigger of the function generator used to generate the sinusoidal excitation was used as an additional input to the correlator. The amplitude is normalized to the amplitude in the DC case ($< 1\text{Hz}$ input frequency). To make this clear to the reader we extended the manuscript with the following sentence: “The normalized frequency response of our device is shown in Fig.2a, where the amplitude $|A|(\omega)$ is defined as the normalized amplitude ($|A|(\omega) / |A|(\omega \rightarrow 0) = 1$) of the modulated optical transmission under a sinusoidal actuation voltage with amplitude $\Delta V(\omega)$ (see schematic in the inset of Fig. 2a, and further description in the Supplementary Section 8).”

Another point of attention are the symbols in the schematics. Why is there an RF source in Fig. 2B? In Fig. 2C a circle with + and – is used for the voltage source, but in other figures it is a circle with a straight line. Please use a consistent symbol. Also, in the latter, the arrow is somewhat confusing as this is typically reserved for current sources (although one can sometimes see it for V source too).

We thank the reviewer for highlighting this inconsistency. We have now adopted the American set of symbols across the schematics. The RF source in Fig 2b is now replaced by a controlled voltage source, indicated by the arrow pointing in from the PID controller. The circuits in the Supplementary are adapted to use the American source symbols as well.

Although it is explained in the text, I found Fig. 3 slightly misleading as the, very nice, device rendering does not correspond to the experiment shown in other panels. Please provide schematics of the experiments 3a and 3b and place the “outlook” rendering in a separate figure. It is also mentioned to do HOM with this device, but how will the required tunable delay (typically cms) be generated on the chip?

We added schematics for the experiments in a new Fig. 3c, and moved the outlook rendering to Fig. 4.

Fig. 3c. Sketch outlining the experimental configuration. For a) the lifetime of the QD is either fully measured on-chip or using the commercial SNSPD system. For b) one arm of the HBT setup is on-chip while the second arm is connected to the SNSPD system. The coincidences are then measured using a time-to-digital converter.

We envision the HOM measurement to be performed with two emitters by tuning their spectral overlap (i.e. in frequency), rather than with a single emitter by tuning the delay between two consecutive photons (i.e. in time). As the reviewer suggests, the latter experiment is extremely challenging to perform in an integrated optics platform due to the long tunable delay lines needed. In contrast, frequency HOM between two emitters does not require long delay lines. Instead, it requires independent spectral tunability of the emitters, which is potentially easier to achieve on-chip, e.g. by using MEMS-induced strain as

described in the original text. To clarify this to the reader, we included the following sentence under Discussion, paragraph 4: “A HOM measurement can then be performed by frequency shifting the spectrum of the emitted single photons of one of the emitters with respect to the other one by means of strain [55].”

In conclusion, authors have done very nice demonstration but need more clarification.

We appreciate the recognition of our work by Reviewer# 2 and hope that our answers and changes in the manuscript answer the remaining open questions. We are open for further discussions, should this not be the case.

Reviewer #3:

The manuscript reports on the integration on the same chip of a low-power MEMS with superconducting nanowire single-photon detectors (SNSPD). This configuration could be of benefit to the development of future integrated quantum circuits because the MEMS actuators do not generate (or require) heat and are therefore compatible with the heat-sensitive SNSPDs.

We thank Reviewer #3 for evaluating our work and we respond to the criticism below.

We agree with Reviewer #3 that the low-energy character of MEMS technology is likely very beneficial to future integrated quantum circuits. We think that in the current outlook of the field, SNSPDs are very important candidates for detection of single photons on chip due to the low noise, not offered by any other single photon detection technology.

The MEMS tunable couplers and SNSPDs are not novel components on their own but their combination on the same chip has not been reported before and therefore contains potential novelty.

In particular, the use of the same NbTiN layer to build the actuators the SNSPD, the electrical connections and contact pads is an interesting solution that could find applications in the fabrication of quantum circuits. However, the manuscript does not contain sufficient novelty and impact to meet the standards of Nature Communications for the reasons detailed below.

We agree with Reviewer #3, combining MEMS technology and superconducting single photon detectors on a single chip is challenging, which limited previous previous work to separate demonstrations. Here we overcome all of the previous challenges through novel integration and testing methods to realize several key components for integrated and quantum optics applications. Both technologies have a long history in the field, the co-integration is not obvious and not an easy task as highlighted by the absence of any such work in the field. Specifically, we believe the demonstration of a simple fabrication process, and the stable operation of SNSPDs under close-proximity high-voltage MEMS actuation, both under static and dynamic operation, is of high value to the quantum photonics community. This, together with our demonstration of PIC reconfiguration, high-dynamic range detection, power stabilization, and single-photon measurements, is a significant step towards demonstrating large-scale quantum photonics, and we believe is very relevant for the broad audience of Nature Communications.

Our view is also kindly shared by Reviewer #1 and Reviewer #2, showing great potential and novelty for the technology suitable for publication in Nature Communications.

Figure 3c is just a 'future vision', far from what is demonstrated and discussed in the paper. For example, not all the components are tested together in the same cryostat (the QD emitter is at 5 K, the SNSPDs are

at 100mK in a dilution fridge); the Hanbury Brown and Twiss (HBT) measurement is performed by routing the photons from the source into an off-chip beam splitter; only one arm is coupled into the on-chip detector, while the other arm is coupled into a commercial off-chip detector.

We agree with Reviewer #3 that Figure 3c is only a future vision, to make this separation clearer, we extended Figure 3 through sketches of the experiments conducted on the chip and moved the outlook to Figure 4. The work is centered on integration of MEMS and SNSPD on the same chip with demonstrator experiments to highlight the usefulness of such a combination.

We agree that the beam-splitter was not used for an HBT kind of measurement, and to avoid any such confusion added the measurement setup sketch in Fig. 3c. Even though we used an external detector to compensate for the coupling loss and poor efficiency of one of our detectors (see explanation below), we believe this HBT measurement verifies single-photon detection in our reconfigurable circuit using an integrated on-chip detector.

Fig. 3c. Sketch outlining the experimental configuration. For a) the lifetime of the QD is either fully measured on-chip or using the commercial SNSPD system. For b) one arm of the HBT setup is on-chip while the second arm is connected to the SNSPD system. The coincidences are then measured using a time-to-digital converter.

As such the paper mostly demonstrates the potential to integrate MEMS and SNSPD on the same chip, but the on-chip measurements are only performed on the individual components.

We would like to clarify that most of our measurements are not performed on individual components, but on our integrated device combining MEMS and SNSPD. For example, we use the MEMS splitter to route light in and out of our integrated SNSPDs under static (Fig. 1d) and dynamic (Fig. 2a) operation, and even directly feedback SNSPD counts to the MEMS beamsplitter (Fig. 2b). The only exception being the HBT measurements which used one external SNSPD, which we now further clarify by adding a subfigure, visually describing the setup, in Fig. 3c. Therefore, we do not agree that measurements were only performed on individual components, and we are confident that our demonstration of MEMS tuning in a SNSPD integrated device, including feedback operations, is a valuable addition to the field, and relevant to a broad audience.

We added Figure 3c (now Figure 4) to highlight how MEMS technology together with this direct feedback can be used to implement many control elements in bulk optical setups and reference experiments such as HBT or HOM type of measurement. We agree that it does not represent what is

currently demonstrated in this manuscript, but something that could be demonstrated by building on this technology and combining it with single-photon source integration and tuning reported in the literature.

Another weakness of the paper is that the authors have not succeeded in doing the HBT 2nd order correlation measurement from the QD source with a pair of waveguide integrated SNSPDs. In Fig 3b the black upper trace is with a pair of commercial fibre coupled SNSPDs; the lower blue trace is with one fibre-coupled SNSPD and one waveguide integrated SNSPD. The authors don't give absolute correlation counts or acquisition times, but the lower trace is very noisy and is clearly barely working. So the claim in the conclusion that they did full second order correlation measurements with the waveguide integrated SNSPDs is not true (the abstract doesn't make this claim

We agree with Reviewer #3 that we have not measured the HBT 2nd order correlation measurement fully on chip. We do not want to create this confusion and thank Reviewer #3 to point this out. We adapted the sentence in the Conclusion to read: "Using our device, we performed on-chip lifetime and, **paired with a fiber-coupled commercial detector**, second-order autocorrelation measurements on single-photons from a quantum dot source."

The reason for not measuring an HBT on-chip is two-fold. First the low coupling efficiency to the chip, and second the lower efficiency on the second detector used for the larger power-range detector. After further investigation of the SNSPD part of the integrated device with the help of an SEM, we included a sentence on this in the main text: "**The detectors show different on-chip detection efficiencies (Detector A is 44.6 times more efficient than Detector B) [25], which we attribute to a lithographic error in detector B, which results in a nanowire constriction and lower current densities, and therefore lower detection efficiency in the remaining nanowire (see Fig. S2).**"

Figure S2: SEM images of the SNSPD hairpins. It is visible that Device B has a defect in the bend leading to a low critical current, limiting the detection efficiency of the straight section.

We used Detector A to measure the life-time of the quantum emitter fully on-chip and used a combination of Detector A and a commercial SNSPD to demonstrate the single-photon character of the measured signal.

From a device point of view there are components that need to be improved or optimised to make this device useful for quantum experiments. For example, the grating couplers have very high losses, which lead to a large discrepancy in the $g(2)$ measurements. Also, the detectors exhibit a very large difference in their efficiencies. These figures are far worse than the performance reported in the literature for similar devices. For example, the authors used grating couplers based on the design Zhu et al., which exhibit far lower losses of $<3\text{dB}$

We agree with the Reviewer #3 that our device components on their own are not showing the performances of state of the art devices. However, we believe that the relevance of our work is that we overcame the challenges to combine these two technologies for the first time, and applied them to several proof of concept experiments. We believe this achievement is more important than showing the best possible performance on individual parts of the device.

On the low grating coupler efficiencies, we want to highlight that the 3dB mentioned by the reviewer, is for the direction from the waveguide to free-space optics. In the work of Zhu et al., they note a simulated input efficiency of 15.9% (-8 dB) at 632.8 nm. In a characterization experiment of a device directly looped back, we have measured 20 dB round-trip loss. In our setup with NA 0.82 (compared to 0.65 in the work by Zhu et al.), we expect a high collection efficiency, so assuming an out-coupling efficiency of 3 dB, we get a coupling loss of 17 dB due to the mode mismatch described in Zhu et al. We attribute the difference to the values reported in Zhu et al. to our sub-optimal geometry combined with our longer operating wavelength and thus smaller mode index step in the grating leading to lower scattering efficiency. We included a discussion in the text on how we designed them and potential ways to improve their efficiency (Results subsection 5, paragraph 3): “The optical coupling efficiency is limited by our grating coupler design, which was experimentally optimized based on the designs in [1]. We believe that our longer wavelength (and thus lower refractive index contrast) and the unoptimized distance to the substrate limits our achievable coupling efficiencies, which can be potentially improved by fabricating additional grating periods.”

Further optimizations based on this superconducting film and the photonic platform are necessary to make the device fully scalable. Many of these optimizations are beyond our local fabrication capabilities (e.g. lack of process stability in a university cleanroom), but are not limiting the technology platform that we demonstrate in this work.

Reviewers' Comments:

Reviewer #1:

Remarks to the Author:

I have read the reply from the authors and the revised manuscript. They have taken proper action on most of my comments, and I think the paper reads better now. However, their response has also created some concern in my judgment about their fabrication process. I still believe that the article deserves some form of publication in Nature Communications; however, proper clarification for the following two points are need.

1- First, the authors track the low yield of the detection down to a physical defect in the second detector. This casts some doubt in my mind about the yield of their fabrication procedure. The authors mention that they fabricated 5 devices, and I am assuming that they present the best one in the paper. Is this the case? If so, this means that the overall efficiency of their fabrication method is lower than 20%, of course with limited statistics. Could you please clarify this?

By the way, do the authors have at least a proper estimate of the efficiency of the first detector? Is it operating upto the expectation from an SNSPD?

2- What is the origin of the defect in detector B? Perhaps the large features of the MEMS switch make the proximity correction harder or is responsible for the defect? A sentence on this would be interesting.

Some further points to make the article more readable:

1- Fig. 1d deserves some explanation about the bottom part of the plot, the log scale extinction. I am unable to understand it. It was my suggestion that the authors plot the extinction in logscale, but I meant to see what is top part of 1d, in log scale.

Reviewer #2:

Remarks to the Author:

With the revision of their manuscript, Gyger and coworkers have addressed my comments and, although there are still a few small points, I can support publication.

- Figure 1d does not show the same quality/layout as rest of the figure. Also, ER is not properly defined and, given that it becomes positive, a nonstandard definition seem have been used. Please define properly
- The normalization is now explained, but please add the data from Fig. 1C in per second to the SI.
- Figure S4: please label the detectors A and B used in the main text
- Please do a careful proofreading of the changed text. e.g. (44.6 ratio, ... does not seem to be correct English.

Reviewer #3:

Remarks to the Author:

In my initial review I stated that the manuscript did not contain sufficient novelty and impact to meet the standards of Nature Communications. I stated the reasons for my decision and the authors' reply has not changed my opinion.

The authors disagree with my comment that the "on-chip measurements are only performed on the individual components". I meant "on-chip QUANTUM measurements are only performed on the individual components". I thought this was obvious considering that the focus of the paper is on integrated chips for quantum applications, and that the word quantum is repeated 10 times in the

abstract alone. However, this should have been made explicit in my comment so I apologise if this was not obvious.

Having clarified this point, I am sure the authors will agree that on-chip quantum measurements were only performed on individual components.

I welcome the changes on the future vision in Figure 3c as it avoids any possible misunderstanding on what was measured on-chip and what wasn't. This change, however, also highlights the fact that little quantum experiments were performed on-chip and, as such, the chip only shows the potential to be used as a quantum chip.

As the authors said in their reply (and I said in my first review), the relevance of their work is that they overcame the challenges to combine the MEMS and SNSPD technologies for the first time. But I disagree when they say "we believe this achievement is more important than showing the best possible performance on individual parts of the device". The main novelty of the paper is about technology and, as it stands, the chip does not show a robust technology that can be used for quantum experiments but only shows some potential, yet to be demonstrated. A credible technological demonstrator would require a substantial improvement in the coupling efficiency, at least a pair of (ideally many more) SNSPDs with comparable performance, a decrease from the current 200V MEMS operating voltage to workable values in the 10s of volts. None of these are trivial problems to solve. And more so, these technological issues become even less trivial to address if they are to be integrated on the same chip with the same technology, which is the main thread of this paper.

Please find below our point-by-point response to the review. The format is as follows: the reviewers' comments are in blue, our responses are in black, with manuscript text indented, and with edited parts in red.

Reviewer #1 (Remarks to the Author):

I have read the reply from the authors and the revised manuscript. They have taken proper action on most of my comments, and I think the paper reads better now. However, their response has also created some concern in my judgment about their fabrication process. I still believe that the article deserves some form of publication in Nature Communications; however, proper clarification for the following two points are need.

We thank Reviewer #1 for appreciating our work and recommending it for publication. We hope the answers below are clarifying the remaining points.

1- First, the authors track the low yield of the detection down to a physical defect in the second detector. This casts some doubt in my mind about the yield of their fabrication procedure. The authors mention that they fabricated 5 devices, and I am assuming that they present the best one in the paper. Is this the case? If so, this means that the overall efficiency of their fabrication method is lower than 20%, of course with limited statistics. Could you please clarify this?

We agree with the referee that the yield is still a challenging topic in the fabrication of SNSPDs. In the last iteration of this manuscript, we added the measurements on 5 SNSPDs, but we notice now that the labeling can be confusing. The measurements are taken from 5 individual pre-characterized SNSPDs on the same chip (in a cleaved portion that was not the one loaded in the dilution refrigerator). As such, all of the SNSPDs measured are identical in design and fabrication as the devices in the manuscript, which also includes the coupling waveguides. In particular, the green and orange curves in Fig. S4 are from SNSPDs with MEMS actuators, similar to the one we focus on in the main text.

Based on our individual SNSPD measurements, combined with our reported two-SNSPD device, we overall observed 7/7 functional SNSPDs. In terms of saturation, we observed 4 saturated SNSPDs (3 from individual SNSPD measurements plus Detector A) with $I_c > 15 \mu\text{A}$ and thus potentially near-unity efficiency, 2 non-saturating devices with $I_c > 10 \mu\text{A}$ with potential for average performance, and 1 device with $I_c \sim 5 \mu\text{A}$ saturating due to constriction (Detector B). From this, we can estimate a saturation yield for SNSPD (counting the constricted device as non-saturating) in the order of $4/7 = 57\%$. Using these limited statistics and compound yield for a two-detector device such as the one central to this work, and given the high yield of the MEMS fabrication (all devices characterized were working), we expect a yield of $\sim 32\%$ for a two-detector device.

To make this clear to the reader and correct this confusion we adapted the sentences in the Supplementary Information to:

Previous to our MEMS measurements, we characterized SNSPDs **from a cleaved portion of the same chip** in a closed-cycle cryostat at 2.5 K, which yielded saturation of internal quantum efficiency at 850 nm for 3 out of 5 tested devices. **All of the detectors are identical in SNSPD design, including waveguide integration, while the green and orange curves are from SNSPDs with MEMS actuators similar to the device reported in the main text.**

and adapted the figure caption of S4 to reflect this.

By the way, do the authors have at least a proper estimate of the efficiency of the first detector? Is it operating upto the expectation from an SNSPD?

From the design of our SNSPD over the length of 20 μm , based on simulations we absorb 95.5% (-13.5 dB) of the light (see Supplementary section 3). We do not expect significant scattering losses in the waveguides on these length scales, as there are no visible defects in the SEM image of the device.

The photon count rate (PCR) versus bias current measurements show a plateau that we verified not being limited by countrate. This is, as commonly accepted by the community, a sign for 100% internal detection efficiency by the detector.

We measured a system detection efficiency of 0.04% (-33.7 dB) which includes coupling to the integrated chip and losses within the tunable beam splitter. A separation of the losses is unfortunately not possible.

We calculated the critical current density of detector A to be $19.5 \cdot 10^9 \text{ A/m}^2$. We can compare this value to critical current densities from our previous work on SiO_2 . Zichi et al. [R1] reports a critical current density of $20.79 \cdot 10^9 \text{ A/m}^2$ with different sputtering conditions to our work but similar Nb/Ti ratio measured at 2.5 K. Gourges et al. [R2], with the same sputtering conditions to our work, reports lower values of $17.4 \cdot 10^9 \text{ A/m}^2$ for the extreme values and $11.3 \cdot 10^9 \text{ A/m}^2$ as average value at 4.2 K. Our device B, which contains the lithography error, shows $7.3 \cdot 10^9 \text{ A/m}^2$ in the straight section. From SEM measurements we estimate a width of 35 to 40 nm at the constriction, which leads to a current density of $18.7 \cdot 10^9 \text{ A/m}^2$ to $16.8 \cdot 10^9 \text{ A/m}^2$, similar to detector A.

Our PCR curve and critical current density, both in line with the state-of-the-art, combined with our demonstrated detection of pure single photons, make us confident that Detector A works up to expectation.

To make these values available to the reader we expanded supplementary section 5 as follows:

From the power meter measurements, we calculate a system detection efficiency from the cryostat window to the detector of -33.7dB for detector A and -49.3 dB for detector B, when all the power is routed to the respective detectors.

[...]

The critical currents (current densities) in the straight sections are 15.8 μA ($19.5 * 10^9 \text{ A/m}^2$) for Detector A and 5.9 μA ($7.3 * 10^9 \text{ A/m}^2$) for Detector B.

The critical current of Detector B is limited by the defect visible in Fig. S2 with an estimated critical current density in the constriction (width of 35 nm) of $18.7 * 10^9 \text{ A/m}^2$ similar to Detector A in the straight section.

2- What is the origin of the defect in detector B? Perhaps the large features of the MEMS switch make the proximity correction harder or is responsible for the defect? A sentence on this would be interesting.

We believe the cause of the constriction is a lithography error. On the same chip, we have a number of devices with different types of MEMS actuators in closer proximity to SNSPDs, and no proximity effect can be observed. To clarify this to the reader, we added the following sentence to the SI:

We observed no constrictions in other SNSPDs in closer proximity to other MEMS structures on the same chip, and thus we attribute this defect to a lithography error.

Some further points to make the article more readable:

1- Fig. 1d deserves some explanation about the bottom part of the plot, the log scale extinction. I am unable to understand it. It was my suggestion that the authors plot the extinction in logscale, but I meant to see what is top part of 1d, in log scale.

We apologize for the misunderstanding. To clarify what was plotted, we explained the plotted parameter in the text and the figure caption, and re-named it as power ratio between detectors (PR):

The measured tuning curve is shown in Fig. 1d, and follows our simulations (see Supplementary section 6), yielding a high extinction ratio of 28.1 dB. The lower part of Fig. 1d shows the power ratio (defined as $PR = 10 \log_{10} \frac{\text{counts}_B}{\text{counts}_A}$) between detectors, and highlights the actuation voltage at which the highest extinction ratio was achieved. We observe an on-off ratio for individual ports of 27.5 dB (detector A) and 11.7 dB (detector B).

And we want to clarify that, based on that definition, positive values indicate that the counts in detector B are higher than in detector A.

Below we plot the requested log scale version of the top part of Fig. 1d. And we want to note that the original data repository is linked to in the manuscript. We decided not to include it in the Supplementary due to the data being redundant, since it forms part of Fig. 1d (both top and bottom) and Fig. S7. If the reviewer feels strongly about including the figure in the manuscript, we will gladly do so.

- [R1] Zichi, Julien, Jin Chang, Stephan Steinhauer, Kristina von Fieandt, Johannes W. N. Los, Gijs Visser, Nima Kalhor, et al. "Optimizing the Stoichiometry of Ultrathin NbTiN Films for High-Performance Superconducting Nanowire Single-Photon Detectors." *Optics Express* 27, no. 19 (September 16, 2019): 26579–87. <https://doi.org/10.1364/OE.27.026579>.
- [R2] Gourgues, Ronan, Iman Esmail Zadeh, Ali W. Elshaari, Gabriele Bulgarini, Johannes W. N. Los, Julien Zichi, Dan Dalacu, Philip J. Poole, Sander N. Dorenbos, and Val Zwiller. "Controlled Integration of Selected Detectors and Emitters in Photonic Integrated Circuits." *Optics Express* 27, no. 3 (February 4, 2019): 3710–16. <https://doi.org/10.1364/OE.27.003710>.

Reviewer #2 (Remarks to the Author):

With the revision of their manuscript, Gyger and coworkers have addressed my comments and, although there are still a few small points, I can support publication.

- Figure 1d does not show the same quality/layout as rest of the figure.

We thank the reviewer for noticing. We have now re-formatted the figures to fit the rest of the manuscript.

Also, ER is not properly defined and, given that it becomes positive, a nonstandard definition seem have been used. Please define properly

We apologize for the misunderstanding. To clarify what was plotted, we explained the plotted parameter in the text and the figure caption, and re-named it as power ratio between detectors (PR):

The measured tuning curve is shown in Fig. 1d, and follows our simulations (see Supplementary section 6), yielding a high extinction ratio of 28.1 dB. The lower part of Fig. 1d shows the power ratio (defined as $PR = 10 \log_{10} \frac{\text{counts}_B}{\text{counts}_A}$) between detectors, and highlights the actuation voltage at which the highest extinction ratio was achieved. We observe an on-off ratio for individual ports of 27.5 dB (detector A) and 11.7 dB (detector B).

And we want to clarify that, based on that definition, positive values indicate that the counts in detector B are higher than in detector A.

- The normalization is now explained, but please add the data from Fig. 1C in per second to the SI.

We extended the sentence in the supplementary to make clear that the given values are per second:

The absolute detector counts in Fig. 1c are 190472 counts per second (Detector A), and 45955 counts per second (Detector B) for a DCR of 100 counts per second.

- Figure S4: please label the detectors A and B used in the main text

In the last iteration of this manuscript, we added the measurements on 5 SNSPDs, but we notice now that the labeling might have been confusing. The measurements are 5 individual SNSPD pre-characterized in the same chip (in a cleaved portion that was not the one loaded in the dilution refrigerator), as such we can not label the ones used in the main text. We did extend the manuscript to make this clear:

Previous to our MEMS measurements, we characterized SNSPDs from a cleaved portion of the same chip in a closed-cycle cryostat at 2.5 K, which yielded saturation of internal quantum efficiency at 850 nm for 3 out of 5 tested devices. All of the detectors are identical in SNSPD design, including waveguide integration, while the green and orange curves are from SNSPDs with MEMS actuators similar to the device reported in the main text.

- Please do a careful proofreading of the changed text. e.g. (44.6 ratio, ... does not seem to be correct English.

We corrected the sentence to:

44.6 times higher in A

We also corrected the following sentences:

... and find a description of the sample and fabrication process in Supplementary sections 1 and 2

... and find further description in the Supplementary section 8

Reviewer #3 (Remarks to the Author):

In my initial review I stated that the manuscript did not contain sufficient novelty and impact to meet the standards of Nature Communications. I stated the reasons for my decision and the authors' reply has not changed my opinion.

The authors disagree with my comment that the "on-chip measurements are only performed on the individual components". I meant "on-chip QUANTUM measurements are only performed on the individual components". I thought this was obvious considering that the focus of the paper is on integrated chips for quantum applications, and that the word quantum is repeated 10 times in

the abstract alone. However, this should have been made explicit in my comment so I apologise if this was not obvious.

Having clarified this point, I am sure the authors will agree that on-chip quantum measurements were only performed on individual components.

I welcome the changes on the future vision in Figure 3c as it avoids any possible misunderstanding on what was measured on-chip and what wasn't. This change, however, also highlights the fact that little quantum experiments were performed on-chip and, as such, the chip only shows the potential to be used as a quantum chip.

Thank you for the clarification of your question. We agree that the measurements with triggered single photons were only performed using a static configuration of the beam splitter to use the high detection efficiency detector. We would like to further highlight that, even though we do not actuate while doing the lifetime or HBT measurements, we do run the on-demand single photons through the waveguides, and we use our waveguide-coupled SNSPD for measuring.

Additionally, we would like to highlight that, in our understanding, quantum photonics also includes the classical components required to build quantum technologies, as long as they adhere to the requirements imposed by quantum applications.

If the source of concern is our quantum-focused title, we could change the title to "Reconfigurable photonics with on-chip single-photon detectors for quantum technologies" if this helps to lift the confusion.

As the authors said in their reply (and I said in my first review), the relevance of their work is that they overcame the challenges to combine the MEMS and SNSPD technologies for the first time. But I disagree when they say "we believe this achievement is more important than showing the best possible performance on individual parts of the device". The main novelty of the paper is about technology and, as it stands, the chip does not show a robust technology that can be used for quantum experiments but only shows some potential, yet to be demonstrated. A credible technological demonstrator would require a substantial improvement in the coupling efficiency, at least a pair of (ideally many more) SNSPDs with comparable performance, a decrease from the current 200V MEMS operating voltage to workable values in the 10s of volts. None of these are trivial problems to solve. And more so, these technological issues become even less trivial to address if they are to be integrated on the same chip with the same technology, which is the main thread of this paper.

Although we do agree that the current technological challenges need to be addressed in follow-up studies, the suggested improvements are out of reach for our infrastructure and resources in a university cleanroom setting. Thus, in the discussion section we outline a clear pathway towards scalable implementations using relevant references that show that 1) higher efficiency coupling is routinely achieved in SiN photonics, 2) sub-10 V photonic MEMS have been reported with other technologies, and we list several paths towards achieving them in this technology, and 3) repeatable and high-yield SNSPDs are possible with scalable fabrication approaches, which are out of our reach as a university cleanroom.

We believe that our contribution demonstrates the fundamental steps required for merging MEMS technology with large-scale quantum photonics, showing its significant potential as well as current challenges, and is thus of immediate interest to several scientific communities, as was also highlighted by Reviewer #1 and #2.

Reviewers' Comments:

Reviewer #1:

Remarks to the Author:

I read the revised version of the manuscript. I am to some degree convinced by the authors that the fabrication yield of the SNSPDs is not low. I am still confused about the yield of a device, including a MEMS switch and two connected SNSPDs. The authors have remained unclear about this, although my comment #1 from the last report. Regardless, based on the statement that they have measured seven detectors from the same chip and the yield was high, I can agree that the combination with the MEMS structures will likely work with high yield as well. Hence, I can recommend the publication of the article.

Reviewer #1 (Remarks to the Author):

I read the revised version of the manuscript. I am to some degree convinced by the authors that the fabrication yield of the SNSPDs is not low. I am still confused about the yield of a device, including a MEMS switch and two connected SNSPDs. The authors have remained unclear about this, although my comment #1 from the last report. Regardless, based on the statement that they have measured seven detectors from the same chip and the yield was high, I can agree that the combination with the MEMS structures will likely work with high yield as well. Hence, I can recommend the publication of the article.

We thank Reviewer #1 for the recommendation to publish the article.